# Semi-automatic mapping of shallow landslides using free Sentinel-2 images and Google Earth Engine.

Davide Notti. [1], Martina Cignetti. [1], Danilo Godone. [1], Daniele Giordan. [1]

[1] Institute for Geo-Hydrological Protection (IRPI), Italian National Research Council (CNR), Torino, Strada Delle Cacce 73, 10135, Italy

*Correspondence to*: Danilo Godone (danilo.godone@irpi.cnr.it)

**Abstract.** The global availability of Sentinel-2 data and the widespread coverage of free-cost and high-resolution images nowadays give opportunities to map, at low-cost, shallow landslides triggered by extreme events (e.g., rainfall, earthquake). Rapid and low-cost shallow landslides mapping could improve damage estimations, susceptibility models and land management.

This work presents a two-phase procedure to detect and map shallow landslides. The first is a semi-automatic methodology allowing for mapping potential shallow landslides (PL) using Sentinel-2 images. The PL aim to detect the most affected areas and focus on them an high-resolution mapping and further investigations. We create a GIS-based and user-friendly methodology to extract PL based on pre- post- event NDVI variation and geomorphological filtering. In the second phase, the semi-automatic inventory was compared with benchmark landslides inventory drawn on high-resolution images. We also used the Google Earth Engine scripts to extract the NDVI time series and make a multi-temporal analysis. We apply this procedure to two study areas in NW Italy, hit in 2016 and 2019 by extreme rainfall events. The results show that the semi-automatic mapping based on Sentinel-2 allows detecting the majority of shallow landslides larger than satellite ground pixel (100 m$^2$). PL density and distribution match well with the benchmark. However, the false positives (30% to 50% of cases) are challenging to filter, especially when they correspond to river bank erosions or cultivated land.

**Keywords:** NDVI, Google Earth Engine, Extreme rainfall event, Free-cost satellite images, Shallow landslides inventory.

## 1. Introduction

One of the recent and forecasted impacts of climate change is the rise of extreme meteorological events (IPCC, 2014). During extreme rainfall, one of the most common phenomena is the activation of shallow landslides (Gariano and Guzzetti, 2016; Guzzetti et al., 2004) as defined by (Guzzetti et al., 2006) and (Caine, 1980). Shallow landslides are not triggered only by rainfall but also by other extreme events like earthquakes (Sassa et al., 1996) or rapid snow melting (Cardinali et al., 2000). These slope instabilities usually involve soils and superficial deposits and represent a meaningful impact on infrastructures (e.g., roads network) and cultivated areas. The pervasive distribution of these phenomena on slopes, hereafter mentioned as extreme events, makes their identification and mapping crucial for effective damage evaluation. For this reason, the definition of procedures and strategies aimed at mapping shallow landslides has been deeply investigated in the last decades to reach different final goals like: i) the mapping of the full extent of a landslide disaster (Guzzetti et al., 2004); ii) geomorphological and erosion studies (Fiorucci et al., 2011); iii) the validation of susceptibility models (Bordoni et al., 2015; Cignetti et al., 2019; Rossi et al., 2010); iv) the statistical comparison of landslides inventories from different methodologies and sensors (Carrara, 1993; Fiorucci et al., 2018).

Landslide event-inventory maps are commonly implemented using several different methodologies: i) post-event aerial photos analysis and plotting (Cardinali et al., 2000); ii) manual or automatic identification based on the use of high-resolution digital elevation models (DEMs) obtained from airborne LiDAR surveys done after the event (D'Amato Avanzi et al., 2015) (Giordan et al., 2017); iii) traditional geomorphological field surveys (Pepe et al., 2019). In recent years, even satellite images have been used to identify and map shallow landslides (Ghorbanzadeh et al., 2021; Lu et al., 2019; Martha et al., 2010; Mondini et al., 2011; Qin et al., 2018). This recent evolution has been possible thanks to the robust improvement of satellite resolution (sub-metric for most commercial satellites), which nowadays is not so different from aerial images (Fiorucci et al., 2019). Recent studies are mostly based on commercial high-resolution satellite images. The use of these commercial images often requires committed acquisition planning after the event that needs a high cost and limits the use of these systems. For instance, areas with low human or infrastructure presence are often overlooked by authorities that mainly dedicate funds to study more inhabited sectors. The scarcity of resources creates a bias between high-income populated areas and remote areas or developing countries that cannot afford the cost.

In the last years, the Sentinel satellites constellation of the Copernicus program made available medium-high resolution images (about 10 m) both multi-spectral (Sentinel-2) and SAR (Sentinel-1) free of cost, and with a high-frequency revisit. In addition, several areas of the world are covered by multi-temporal very-high-resolution images of GoogleEarth™ that could help to detect and map shallow landslides when pre- and post- event images are available (Borrelli et al., 2015). Google Earth Engine (GEE) cloud processing (Gorelick et al., 2017) could also be used to create time series of several satellite data( Optical, SAR), which are useful to detect the change and the recovery of vegetation and to map landslides and their effect on vegetated areas (Scheip and Wegmann, 2021; Yu et al., 2018; Handwerger et al., 2022; Lindsay et al., 2022; Ganerød et al., 2023).

In this study, we utilized pre- and post-event NDVI (Normalized Difference Vegetation Index) data from Sentinel-2 to develop a dedicated methodology for semi-automatically detecting potential shallow landslides. To assess the accuracy of our approach, we compared the potential landslides detected using our method with a benchmark inventory manually mapped on post-event high-resolution images.

We utilized the Google Earth Engine (GEE) platform to generate NDVI time series, which allowed us to pinpoint the optimal image pairing for detecting potential landslides, calculate multi-temporal NDVI averages, and keep track of vegetation regrowth in the impacted regions.

Our methodology aims to provide a more user-friendly approach compared to similar studies. We achieved this by using free-cost data, open-source software, and empirical thresholding, which makes it easier to replicate our approach in other regions affected by shallow landslides. The implemented methodology has been tested in two areas of north-western Italy hit by extreme rainfall events in recent years, *i.e.,* November 2016 and October 2019. The two events triggered hundreds of shallow landslides in small areas, causing widespread damage to the road network, cultivation and, in some cases, urban areas.

Semi-automatic and manual inventories and GEE scripts are also published online and open for improvement by the scientific and user community.

## 2. Study areas

The two presented case studies are located in NW Italy, respectively, affected by two heavy rainfall events in November 2016 and October 2019.

The 2016 event area (about 350 km$^2$) is located in the Ligurian Alps at the border between Liguria and Piemonte regions (NW Italy). The shape and the extension of the study area (Figure 1) are a combination of: i) the area most hit by rainfall, ii) other literature studies of the event (Cremonini and Tiranti, 2018; Pepe et al., 2019), iii) the footprint of the available Sentinel-2 cloud-free images and the post-event Google Earth image. The area of interest (AOI), henceforth called Tanarello and Arroscia valleys, shows an elevation up to 2500 m a.s.l., and a wide range of land use and vegetation cover from the Mediterranean to the alpine environment. The area intersects several river basins, and the main catchments are the Tanaro-Tanerello, part of the Po river Basin and the Arroscia stream flowing to the Ligurian Sea. From the geological point of view (Figure 1 B), the northern sector of the area is occupied by the Briançonnaise Zone of the middle Pennidic nappe. This unit is represented by limestone-dolomite, which creates steep slopes, conglomerate and volcanic formation (rhyolite). In the southern part of the area, outcropping the Helminotod flysch formations of Monte Saccarello-San Remo, made by a limestone-clay sequence, and the sandstone-siltstone sequence of San Bartolomeo formation (Lanteaume et al., 1990; Pepe et al., 2015). The Tanarello and Arroscia valleys area has a sparse human settlement and low population density, ranging from 40 to 1 inhabitant per square kilometres. Most of the inhabitants live in Ormea and Pieve di Teco towns. Most of the area is occupied by broadleaf forests in the lower part and coniferous forest, grassland and pasture at high altitudes.

The area affected by the heavy rainfall event in 2019 (about 530 km$^2$) is located between the Bormida river and Lemme valleys, in the Southern-east Piemonte region. The considered area has been delimited considering the effects of the event based on the rainfall data, image coverage and reports on damages. The study area mostly overlaps with the Tertiary Piedmont Basin (TPB): a sedimentary succession from Oligocene conglomerates in the South to Pliocene mudstone in the Northern part (Figure 2 B). Three main geological formations outcrop in the detailed training areas, from South to North: the Cortemillia formation (made by Arenite, Mudstone); the Cessole Marls (made by carbonate-rich mudstone, arenite); the Serravalle Formation (made by arenite and sandstone). The southern part is occupied by ophiolitic rocks of the Ligurian oceanic unit (Piana et al., 2017). Alluvial quaternary deposits occupy the bottom of the valley. Several small creeks cross the study area with S-N directions that, in the 2019 event, caused flash floods (Mandarino et al., 2021). The presence of a gentle hilly landscape characterizes the geomorphology of the area. However, the slope is steeper in the northern sector than in the rest of the training area, where the Serravalle Formation outcrops. The vineyards (region of Gavi grape) are mainly located in the central and southern portions of the study area (Cessole Marl formations). In contrast, the northern-western part is mainly covered by broadleaf forest, sclerophyllous vegetation and shrubs. Several villages and the small town of Gavi are located inside the AOI, henceforth called Gavi area. Shallow landslides frequently hit the Castle of Gavi hill, like in 2014, 1977 and 1935 events (Govi, 1978; Mandarino et al., 2021).

From the climatological point of view, (Fratianni and Acquaotta, 2017) Tanarello/Arroscia is between the Alpine and Liguria- Tyrrenian climate area, while the Gavi area is is between Po Plain and Upper Adriatic Region and Alpine and Liguria- Tyrrenian zone. Moreover, the area of Gavi is also close to the area with high-frequency of intense rainfall (Fratianni and Acquaotta, 2017) in the NE sector of Liguria.

Recently, global warming and the related sea temperature increase caused a likely positive trend of extreme rainfall events in the area of the Ligurian Sea, especially on a short time interval (i.e. <24 hours) (Gallus Jr et al., 2018; Paliaga and Parodi, 2022; Roccati et al., 2020).

**2.1 The 20-25 November 2016 event in Tanarrello and Arroscia valleys.**

Historical information tells us that the Liguria region and NW Alps have been usually affected by several extreme rainfall events, usually during autumn (D'Amato Avanzi et al., 2015; Cevasco et al., 2014; Ferrari et al., 2021; Roccati et al., 2018; Guzzetti et al., 2004; Luino, 1999). From 20 to 25 November 2016, a low-pressure area affected the western Mediterranean Sea (Nimbus Web Eventi Meteorologici, 2022), causing heavy and persistent rainfall that hit NW Italy, and with high severity, the Ligurian Alps. The upper valleys of Tanarello and Arroscia streams (at the border between Liguria and Piemonte regions) were the most hit by this event, and the rainfall accumulation reached 650 – 700 mm (Figure 1 C). The rain gauge station of Piaggia reached a value of 690 mm over five days, which is far higher than previous extreme events of the last 70 years (Figure 1 D). The dense network of rainfall gauges of ARPA Piemonte and ARPA Liguria (the regional agencies for environmental protection) allowed to create a 1 km spatial resolution map of accumulated rainfall and compare precipitation time series for some stations. This heavy rainfall triggered many shallow landslides partly mapped with field surveys in Arroscia valley (Pepe et al., 2019). Also, deeper landslides were triggered, like in the case of the villages Monesi di Mendatica, which were partly destroyed by such kind of landslide (ARPA Piemonte, 2018; Notti et al., 2021). Despite the limited human presence, the damages are estimated in several Millions of Euros only for public infrastructures.

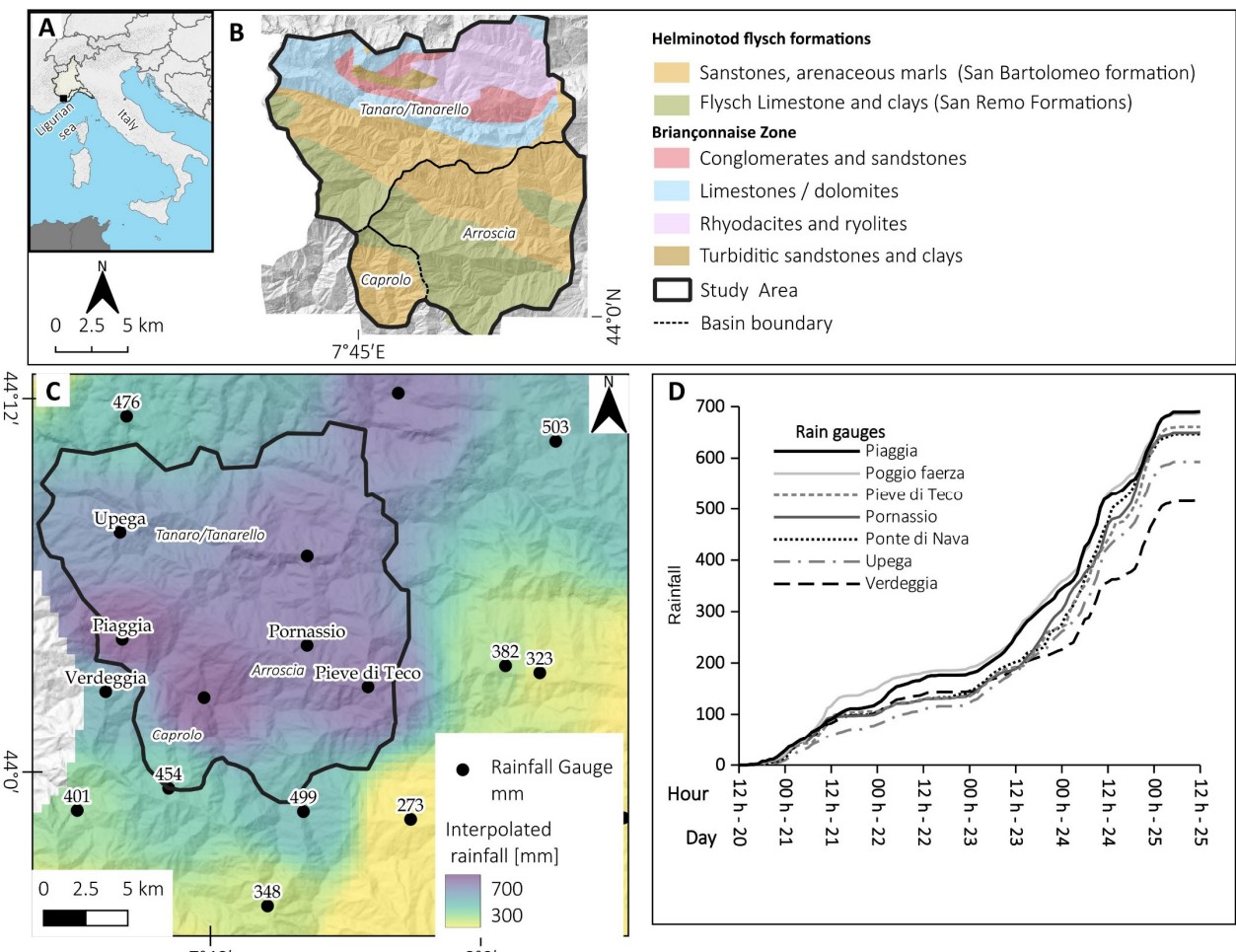

**Figure 1. Arroscia/Tanarello November 2016 event. A) Location of the study area; B) Simplified geological map based on (Lanteaume et al., 1990); C) Accumulated rainfall from 20 to 25 November 2016 in the study area. D) Hourly cumulated rainfall for some rain gauge stations of study areas. (Rainfall source: ARPA Piemonte and ARPAL Liguria). Shaded reliefs of maps B and C are based on the DTM of ARPA Piemonte and Regione Liguria.**

**2.2 the 21-22 October 2019 event in Gavi area.**

The October to December 2019 period has been characterized by numerous meteorological events that hit NW Italy, causing an extreme rainy period (Copernicus Climate Change Service, 2019). In particular, on 19-21 October 2019, an extreme rainfall hit an area between Liguria and Piemonte regions, causing severe floods and diffuse shallow landslides in the basins of the Orba and Bormida rivers (Mandarino et al., 2021). This event was caused by a semi-stationary V-shape storm over a relatively small area with extreme rainfall (Figure 2 C) both in hourly intensity and total accumulation

(Mercalli, 2019). This event activated many shallow and deep landslides. In particular, we focused on the area near the town of Gavi where the rain gauge registered about 480 mm/24h, and most of the rainfall (318 mm) was concentrated in six hours intervals (Figure 2 D) (Meteologix, 2022). It is one of the highest rainfall records in the Piemonte region, just five years before (October 2014), extreme rainfall hit the same area. The extreme rainfall events of autumn 2019 caused estimated damages of 16 M of Euros in the province of Alessandria (Regione Piemonte - flood events 2019, 2021). After

the October 2019 event, a particularly wet period triggered other shallow landslides in the study area until December 2019.

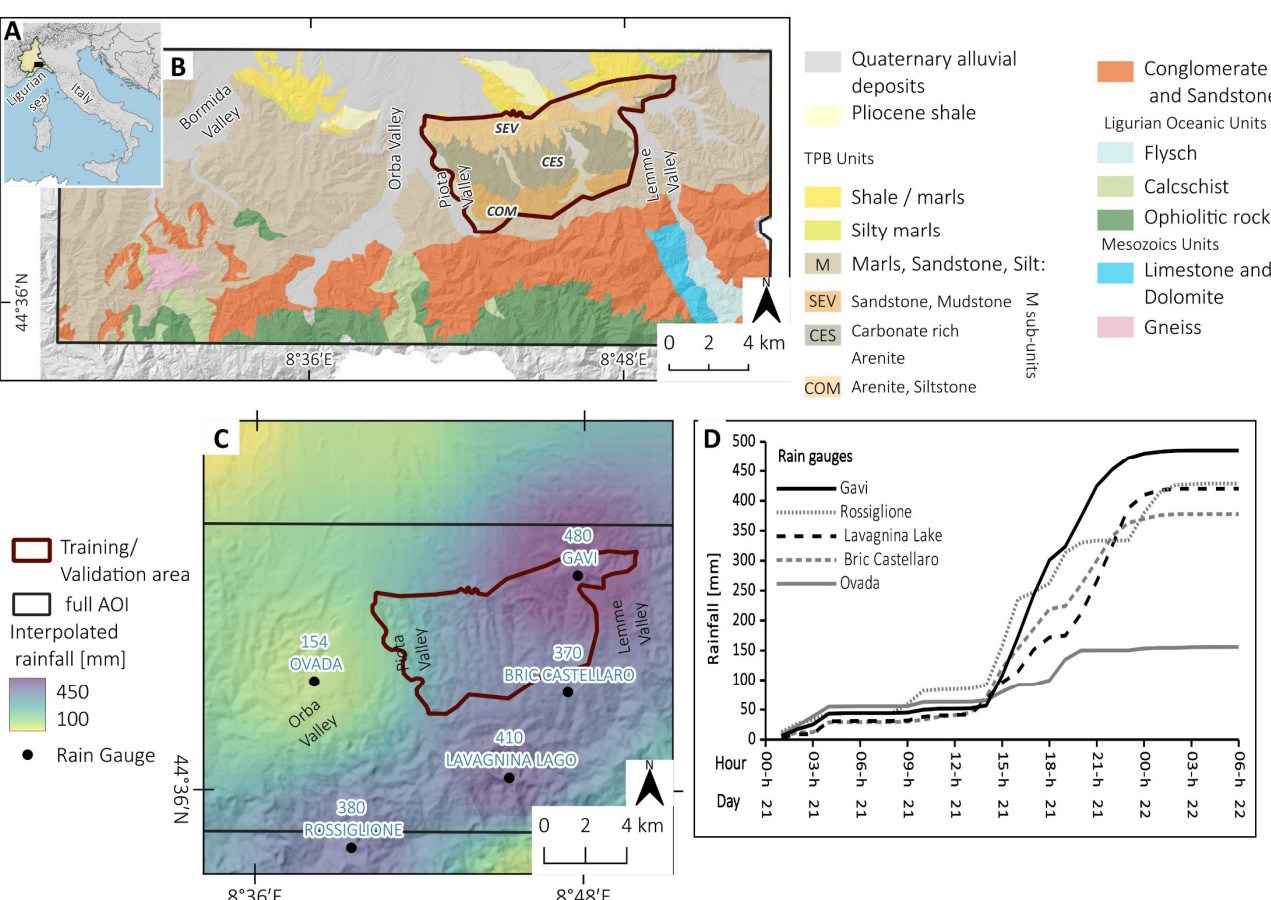

Figure 2. Gavi October 2019 Event. A) Location of the study area; B) Simplified lithological map of the study area based on (Piana et al., 2017); C) Accumulated rainfall from 21 to 22 October 2019 event. D) Hourly cumulated rainfall for some rain

gauge stations of study areas. (Rainfall eata source: ARPA Piemonte). Shaded reliefs of maps B and C are based on the DTM of ARPA Piemonte and Regione Liguria.

**3. Materials and Methods.**

One of the main effects caused by the activation of shallow landslides is the reduction of vegetation cover that creates a radiometric signature variation often detected by multispectral satellites. Thus, the NDVI is one of the most used band

indexes to detect these variations (Fiorucci et al., 2019; Lu et al., 2019; Mondini et al., 2011). For this reason, we create

a semi-automatic methodology that deals with pre- and post-event NDVI variation based on free satellite images with the best spatial resolution (Sentinel-2, nowadays).

Our methodology, resumed in Figure 3, aims to produce an inventory of potential shallow landslides (PL) based on NDVI and geomorphological filters. The proposed method has two main phases: i) in the first one, a semi-automatic methodology, divided into 3 main steps, identifies the potential high-density shallow landslide areas; ii) In second phase and then the semi-automatic inventory will be further used to support the manual mapping of landslides on very high-resolution images. Time series of NDVI computed on GEE were also used to evaluate the vegetation recovery.

### 3.1 Potential landslides detection methodology

The proposed methodology (Figure 3) is aimed to create a PL inventory based on the semi-automatic procedure. The mapping method is based on the availability of pre- and post-event moderate resolution (10 m) satellite images. This methodology is intended to detect surface changes, which are signs of potential shallow landslides, between pre- and post-event images. The PL inventory is aimed to delimit the area most affected by shallow landslides and support a subsequent, detailed landslide mapping on very high-resolution (< 1 m) images (satellite or aerial based). The PL is not a geomorphological landslide inventory because the shape of PL is extracted with a semi-automatic procedure and based on middle-resolution images. The PL inventory is created in three main steps: i) satellite images selection; ii) calculation of the Normalised Difference Vegetation Index variation ($NDVI_{var}$), and definition of empirical $NDVI_{var}$ threshold that is adopted for the potential landslide mapping; iii) implementation of a filter using terrain and other geomorphological properties slope to obtain the potential landslide (PL) inventory and the PL density maps. Thus, the PL inventory is compared over a training area covered by high-resolution images, with a manually drawn dataset, i.e., manual landslides (ML). This comparison phase is important because is used for checking the efficiency of PL methodology and refining the calibration of adopted parameters with iteration processes to improve the quality of the final PL inventory, reducing errors. The proposed methodology is exclusively based on free-cost software (*e.g.* QGIS (QGIS Association., 2022), SAGA GIS (Conrad et al., 2015), R(R Core Team, 2020)) and cloud computing (*e.g.* GEE). In the following sections, the procedure is discussed in more detail.

### 3.1.1 Satellite image selection.
The first step is the selection of the best pairs of pre- and post-event satellite images aimed at making a change detection analysis. Nowadays, the free satellite images with the best resolution are Sentinel-2 (10 m visible and near-infrared bands) followed by Landsat ones (30 m).

We search for images and filter images using the following criteria:

    i.     cloud cover < 5%;

    ii.    images acquired in the same period of the year to minimise the effect of shadow and canopy cover change related to season;

    iii.   the season with the highest NDVI, to obtain a strong contrast in NDVI, no snow coverage, and short shadow. The constraint period depends on local climatological conditions (e.g., summer from June to September in the middle latitude of the N Emisphere).

To improve the search for the best pair of images, we also used output from a GEE processing based on the code developed by (Nowak et al., 2021). The processing calculates a temporally averaged NDVI time series from a satellite images

collection (e.g. Sentinel-2) filtered by cloud cover (< 5 %, or less if possible) over selected sample polygons that can be directly drawn on the satellite map interface of GEE. The time series plotted in a GEE chart can be exported to a CSV file for further filtering (e.g. replicate date removal) and analysis. We obtained a limited number of pairs of pre- and post-event images with these constraints.

### 3.1.2 NDVI$_{var}$ calculation and threshold

In the second step, we calculated NDVI$_{var}$ by computing the NDVI variation between the pre- and post-event conditions (equation 1). The aim is to identify areas with decreased NDVI values due to vegetation removal or damage caused by shallow landslides. Using the raster calculator of QGIS software, we computed the NDVI using the NIR and the red band. In the specific case of our study areas, we used the Sentinel-2 band of NIR (Band 8) and red (Band 4) which have a spatial resolution of 10 m.

$$NDVI_{VAR} = \ NDVI_{post} - NDVI_{pre} \ \text{Where NDVI} = (NIR - Red)/(NIR + Red) \tag{1}$$

Then we manually select NDVI$_{var}$ threshold that best identifies changes related to landslides. This threshold does not have a fixed value, as reported in the literature (Hölbling et al., 2015; Mondini et al., 2011). An operator manually determines it after visual assessment of NDVI$_{var}$, the comparison, on GEE, of NDVI times series between affected/not affected area, and the calibration of the parameters based on PL/ML inventories comparison (back analysis). For instance, in our study areas, both the visual pattern of NDVI$_{var}$ and the observation of NDVI time series suggest that the optimal threshold should be in the range of -0.20 / -0.15, but this could be different for other cases.

### 3.1.3 Geomorphological filtering to create PL inventory.

In the third step, the results coming from NDVI$_{var}$ are filtered using geomorphological parameters (Table 2). We first used the slope derived from DTM (download from Regione Piemonte and Liguria databases) to filter out the areas with a slope angle below a certain threshold. Also, in this case, the value is empirically based on the visual pattern and back analysis (slope distribution of ML). Specifically, to create PL, we applied this procedure: i) the rasters of NDVI$_{var}$ and slope are converted in a boolean raster (0-1) using the thresholds mentioned above (e.g., B = [NDVI$_{var}$<-0.16 and slope>15°]) in a raster calculator of QGIS, in the computed raster the value 1 corresponds to the potential landslides (PL); ii) Then on QGIS, we converted the value 1 of the raster into vectors (polygon); iii) the median slope (always with QGIS) is calculated for each PL polygon and further filtered with a certain threshold (e.g. > 17°); iv) the polygons are smoothed to obtain a more geomorphological shape. Additional filters may be introduced based on radiometric (e.g., removing the area in permanent shadow) or geometric parameters (e.g., removing the PL that overlaps with a riverbed). These filters based on empirical thresholds should be evaluated case by case considering the morphology, the land use and the ancillary data available in the study area. We finally obtained the final PL inventory with their centroid.

We used the PL centroid to create with QGIS a Kernel (Terrell and Scott, 1992) heatmap relative density (KD) map of relative PL. The PL KD aims to identify the most affected area. The parameters (e.g., search radius, size of the cell ) used to generate KD maps depend on the dimension of the study area and the average distance of PL centroids. The same procedure is applied to the ML to create a density map for inventory comparison.

### 3.1.4 Parameters calibration based on ML inventory comparison

Usually, the manual mapping is done on a representative training area (e.g. at least 70 % of the whole AOI) (Mondini et al., 2011; Mohan et al., 2021; Trigila et al., 2013), and it is used as a benchmark dataset for calibrating the proposed method, then it is validated on a small area (30 %). In this work, having to test a new methodology, we decided to operate through calibration and manual mapping on 100% of the area of interest to determine if the technique is sufficiently robust.

The first PL inventory is then compared with a ML drawn on high-resolution images in a limited subarea of the AOI. The identification of the training area is based on the following criteria: i) availability of cost-free high-resolution post-event images, ii) PL density map, described in the previous paragraph, iii) intensity of the event (*e.g.,* accumulated rainfall), iv) representativity of the study area in terms of lands-use and geomorphology. Once the result of the training area is available, it is possible to evaluate the statistical distribution of ML inventory in terms of NDVI$_{var}$, slope angle or other parameters for a better empirical calibration of the thresholds used to create the PL. The calibration step aims to reduce false positives and false negatives and obtain the definitive PL inventory.

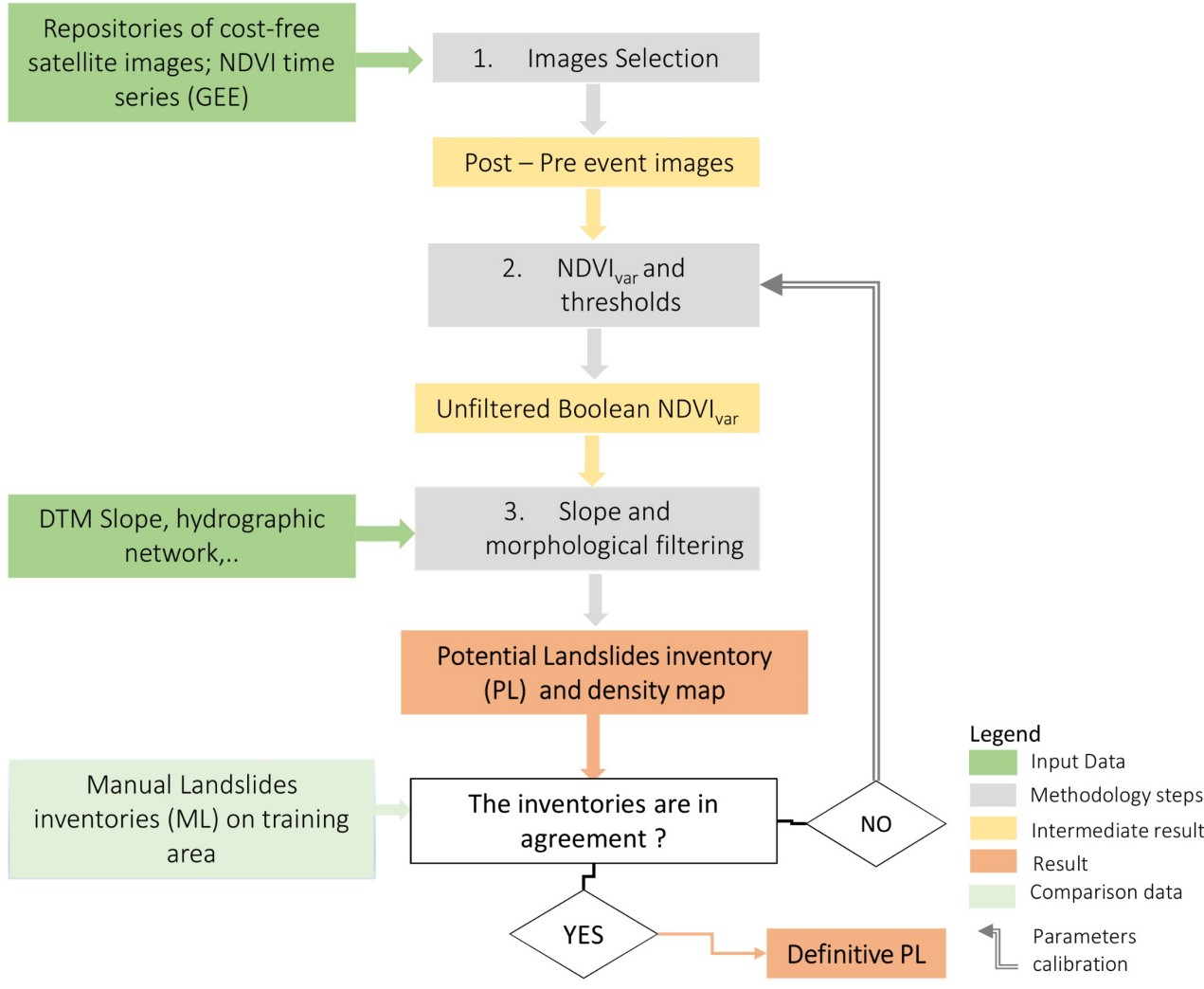

**Figure 3. Flow chart of proposed shallow landslides mapping methodology.**

## 3.2 Data, parameters and thresholds used for the case studies

We applied the methodology described above to our two case studies; for each, we chose the data and parameters

described in the following paragraphs. The data availability section provides the link to the original databases used.

### 3.2.1 Satellite data and geomorphological filters

Following the methodology described in section 3.1, we first selected the pre- and post- images. For our case studies, we used the data of Sentinel-2 satellites (type Bottom-Of-Atmosphere reflectance in cartographic geometry - L2A) available on the Sentinel-Hub portal (Copernicus Open Access Hub, 2022). In particular, we downloaded the 10-m resolution bands

visible (B2, B3, B4) and near-infrared (NIR - B8), and we clipped the original datasets on the area of interest. The dataset used in this study are listed in Table 1. For the Gavi area, we find a pair of images with one year of interval, while for Tanarello and Arroscia valley area, the best pair cloud-free is on two years interval.

In the second step, we defined the NDVIvar and geomorphological thresholds. In our case study, we used $NDVI_{VAR} \leq -0.15$ joined to slope> 15° entries values to create the Boolean raster where the value 1 identifies the potential landslides.

The slope models were computed from 5-m DTM available from the databases of the Piemonte and Liguria regions, and adopted to the spatial resolution of Sentinel-2 in the raster calculator of QGIS (i.e. it averages the values of 4 cells of 5x5 m into one cell 10x10 m). The boolean raster keeps the same spatial resolution of Sentinel-2 (10x10 m pixel).

We used additional geomorphological thresholds, such as the median slope of the PL polygons. The values were based on the statistical distribution of the ML median slope. We used the filter values median slope >= 17° and >= 20° for the

Tanarello and Arroscia valley area and the Gavi area, respectively.

In the Tanarello and Arroscia valleys case, we removed the polygons nearby (using an empiric buffer from 5 to 10 m) the hydrographic network to remove false positives related to riverbank erosion. We also filtered the areas in a constant shadow because of a cliff, using threshold values of the averaged means of the four 10-m bands (RGB and NIR) of Sentinel-2 images. The PL polygons were finally smoothed and merged if their distance was lower than 1 m.


**Table 1: Sentinel-2 images used in the selected case studies**

| Case study | Event date | Image Source | Band | Spatial resolution | Pre-Event images | Post-Event Images | Usage |
|---|---|---|---|---|---|---|---|
| Tanarello Arroscia valleys | 20-26 November 2016 | Sentinel-2 | R-G-B; NIR | 10 m | 23/08/2016 | 28/08/2018 | Semi-Automatic detection |
| | | GEE | NDVI | 10 m | 2016-2021 | | NDVI time series |
| Gavi area | 20-26 October 2019 | Sentinel-2 | R-G-B; NIR | 10 m | 26/06/2019 | 20/06/2020 | Semi-Automatic detection |
| | | GEE | NDVI | 10 m | 2016-2021 | | NDVI time series |

### 3.2.2 High-resolution images for ML inventory.

For our study, we employed a stringent calibration and validation methodology in both of our study areas, with an equal ratio of 1:1. This differs from the commonly used approach in the literature of a 7:3 ratio, as reported in studies such as Mondini et al. (2011) and Mohan et al. (2021). Our decision to utilize this methodology was motivated by the fact that it was our first time implementing this approach. Furthermore, we employed a statistical validation method that is user-friendly and straightforward to replicate in future studies.

For the Tanarello-Arrocia case, the training-validation area is about 300 km$^2$, while it is 50 km$^2$ for Gavi 2019 case. Concerning the Gavi area case study, we applied the proposed methodology as in an ordinary scenario; therefore, we performed, over 10% (about 50 km$^2$) of the area training and validation, and then, with the previously defined parameters, we applied the method to 90% (500 km$^2$) of the area, producing the inventory. The large 2019 area is also intended to be used as a test inventory for other studies.

The manual mapping of the landslides was made (in early 2022) on very high-resolution post-event images. In our case study, high-resolution images were available for both areas, dating back to a few months after the events (Table 2). We used the post-event satellite images available on Google Earth upload as XYZ tiles layer on QGIS software. We also used pre-event high-resolution orthoimages available as a web map service (WMS) on the national cartographic service of Italy or the regional web map service of Piemonte and Liguria regions.

The manual polygons of landslides (ML) were drawn following geomorphological criteria with the help of shaded relief DTM. Two operators manually checked the inventories to reduce the subjectivity of mapping. The landslide mapping was made on QGIS with the support of Google Earth Pro for historical image visualisation.

**Table 2. High-resolution images used for ML mapping in this work.**

| Case study | Image Source | Band | Spatial resolution | Pre-Event images | Post-Event Images | Usage |
|---|---|---|---|---|---|---|
| Tanarello and Arroscia valleys | Google Earth; National cartographic portal (PCN) | Visible | 0.3 m | 24/09/15 2012 | 03/08/17 | PL Validation and Manual mapping |
| Gavi area | Google Earth; Piemonte regional webgis service | Visible | 0.3 m | 01/06/2019 2018 | 07/04/2020 16/03/2021 | PL Validation |


### 3.2.3 NDVI time series analysis using GEE.

    We also used GEE's potentiality to check and compare the NDVI time series affected by shallow landslides. We manually draw on the GEE interface some sample polygons, imported as feature collections representative of different conditions (landslide / no landslides) and land use. As for the choice of the best images described in section 3.1, we used code based
on (Nowak et al., 2021). We also extract the NDVI time series using the GEE time-series explorer QGIS plugin (GEE Timeseries Explorer), such script allows producing single-pixel time series directly from a point vector. The time series analysis aims to estimate vegetation recovery in the area affected by shallow landslides and compare it with healthy areas. The vegetation recovery helps asses the maximum period in which a post-event image can be used to calculate NDVI$_{var}$.
The NDVI time series located in different land use were also compared using GEE.

### 3.3 PL and ML inventories comparison and statistics

    We compared the PL with ML to evaluate the efficiency of semi-automatic detection by using geoprocessing tools in a GIS environment. Results were synthesized in a validation matrix, the graphical sketch in Figure 4 shows all the possible combinations. PL and ML vector layers were merged in a unique one to obtain the intersections between the two datasets,
thus getting true positive (TP), then by selecting residuals PL (i.e. parts not included in the intersection) touching or not the intersection, partial positive (PP) and false positive (FP) were detected. On the other hand, partial detection (PD) and

false negative (FN) were defined, respectively, by residual ML touching the intersection sector and ML not intersecting with PL. The five categories were merged in a unique vector layer for further analysis, with the aforementioned classification stored in the attribute table.

The three intersections involving PL (TP, PP and FP) were used to analyze and validate the semi-automatic methodology. Then, by overlaying them on NDVI and slope angle raster layer, mean values were calculated and stored for each feature. Those values were then processed to obtain descriptive statistics and frequency distributions in terms of the previously described categories. The ML/PL comparison of datasets was also used in an iteration process to enhance the parameters to obtain the PL. The characteristics of FP and TP (e.g. slope and $NDVI_{var}$ distribution) allowed for improving the filters 325 in semi-automatic detection. We applied some equations similar to those commonly used in literature (Prakash et al., 2021; Nava et al., 2022; Catani, 2021) to check the quality of automatic mapping. Thus, the false-positive rate (FPR, equation 2) measures the percentage of the area not correctly detected, and the detection rate (DR, equation 3) is the percentage of shallow landslides (both full and partially) detected by the PL methodology. We also analyse the factors influencing the DR, such as shallow landslide dimension or land use.


$$FPR = (FP) / [(FP) + (TP) + (PP)] \ (2)$$
$$DR = [(TP) + (PD)] / [(TP) + (PD) + (FN)] \ (3)$$

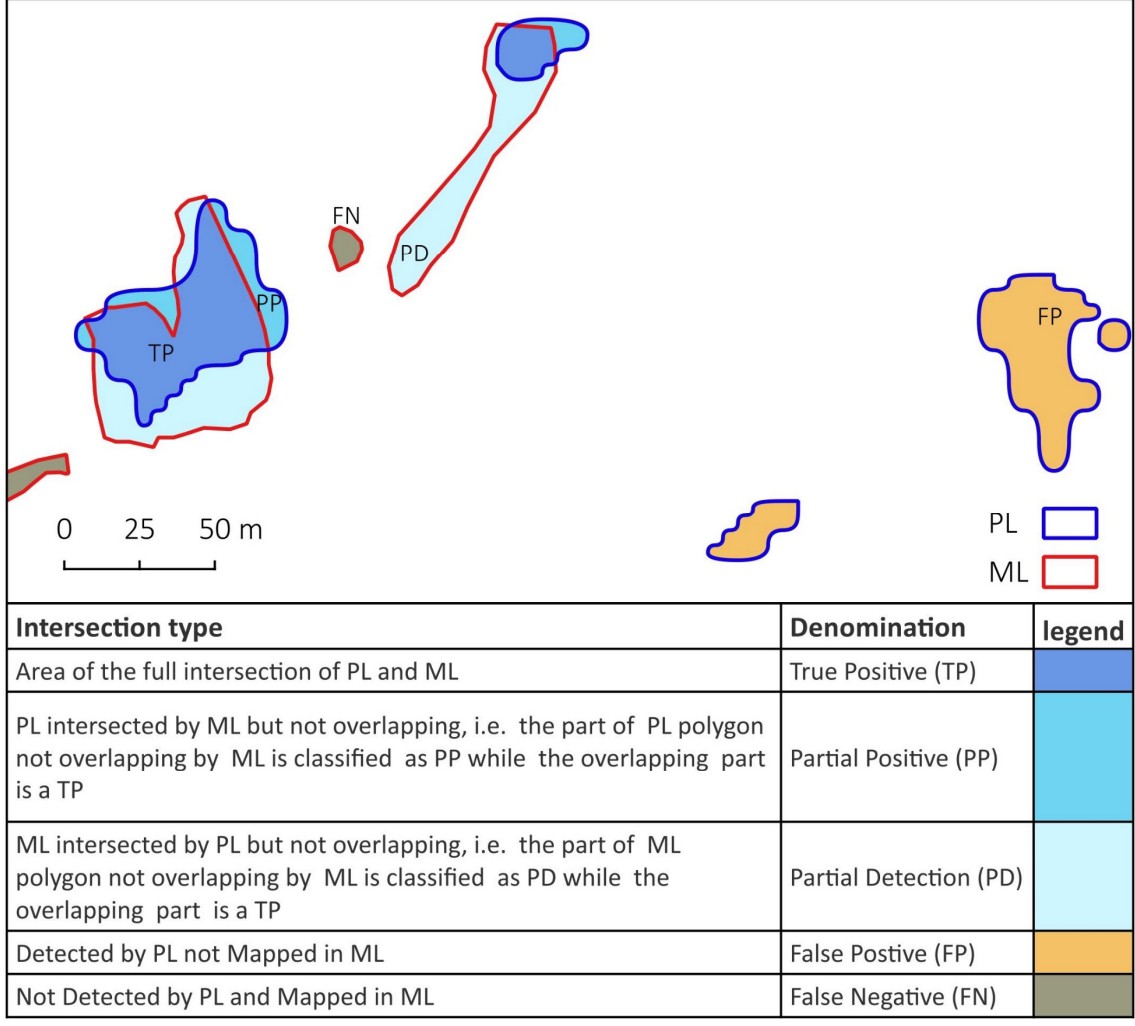

| Intersection type | Denomination | legend |
|---|---|---|
| Area of the full intersection of PL and ML | True Positive (TP) | |
| PL intersected by ML but not overlapping, i.e. the part of PL polygon not overlapping by ML is classified as PP while the overlapping part is a TP | Partial Positive (PP) | |
| ML intersected by PL but not overlapping, i.e. the part of ML polygon not overlapping by ML is classified as PD while the overlapping part is a TP | Partial Detection (PD) | |
| Detected by PL not Mapped in ML | False Postive (FP) | |
| Not Detected by PL and Mapped in ML | False Negative (FN) | |

**Figure 4: The five possible combinations of PL–ML intersection cases with their description**

We also compare the PL and ML inventories using the KD maps made with the procedure described in paragraph 3.1.3. The two densities, sampled on the same regular grid, are compared in a scatter plot, and a correlation coefficient is calculated (Benesty et al., 2009).

 **4. Results and discussion**

The proposed methodology was applied to the two case studies of the Tanarello and Arroscia valleys (2016) and the Gavi area (2019). We obtained for both areas PL and ML inventories with more than 1000 shallow landslides mapped for each AOIs. Statistics about the efficiency of PL are also presented in the following paragraphs. Finally, other ancillary datasets were compared to analyse shallow landslide distribution characteristics.

 **4.1 Tanarello and Arroscia valleys study area**

The semi-automatic mapping based on $NDVI_{var}$ of pre-event (2016-08-23) and post-event (2018-08-28) Sentinel-2 images, for the Tanarello and Arroscia valleys study area provided 1056 PL.

Figure 5 shows an example of the steps and results of our methodology over a sample area of the Arroscia/Tanerello case study. From the comparison of pre- (Figure 5 A), and post-event images (Figure 5 B), we obtained the $NDVI_{var}$ that was  used to extrapolate the PL (Figure 5 C). Figure 5 D compares the ML draw on high-resolution Google Earth satellite images and the PL. A detailed 3-D view from Google Earth Pro of pre- and post-event is shown in Figures 5 E and F.

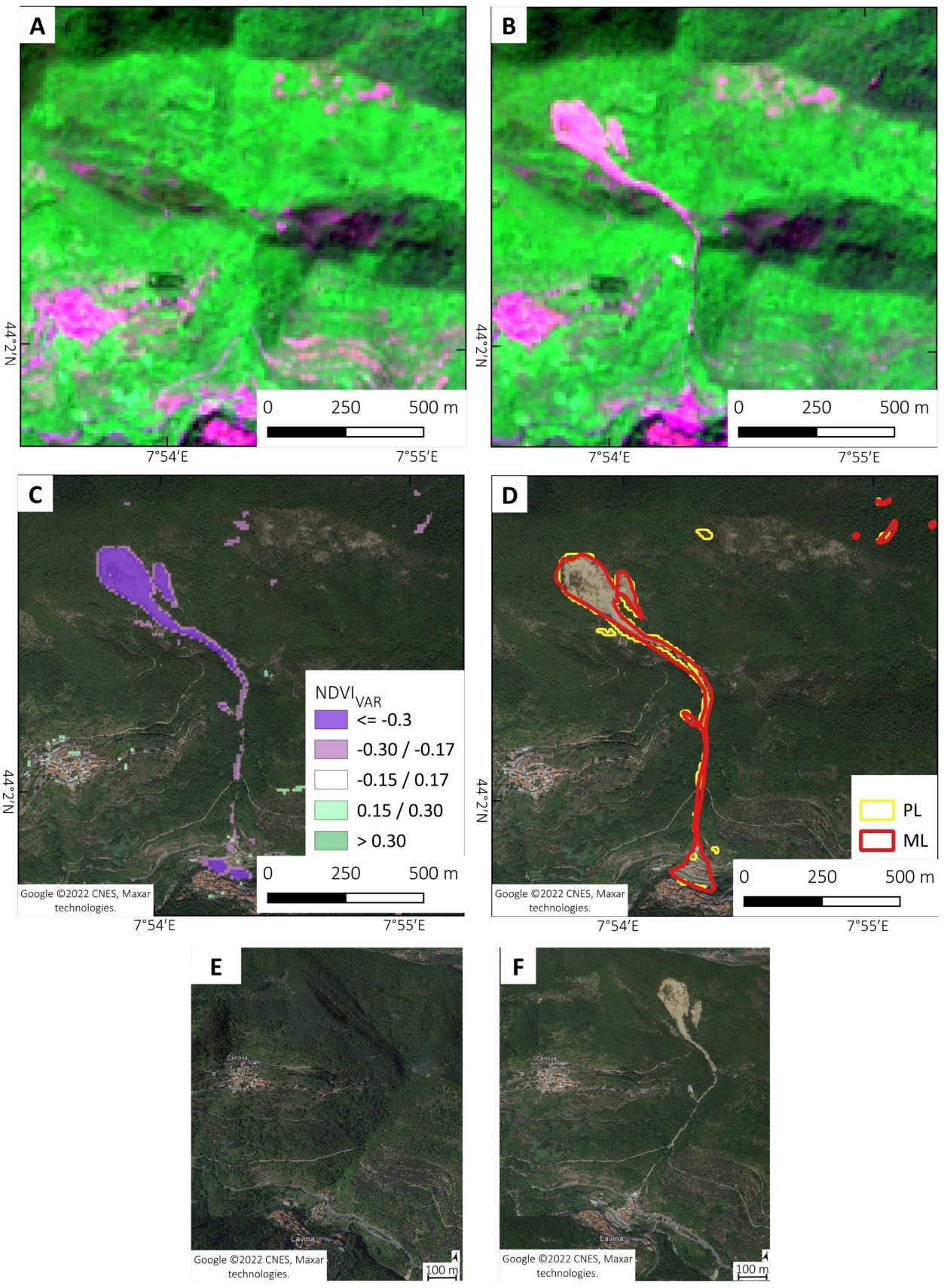

**Figure 5. Example of shallow landslides detection methodology for Tanarello and Arroscia valleys case study: A) Sentinel-2 pre-event images (RED-NIR-BLUE) acquired on 2016-08-23; B) A) Sentinel-2 post-event images (RED-NIR-BLUE) acquired on 2018-08-28; C) NDVI$_{var}$ map PL inventory; D) ML and PL inventory overlapped to post-event high-resolution image (Google Earth – 2017). Google Earth 3D view of pre- (E) and post- (F) events of the area affected by landslides. Maps data: Google ©2022 CNES, Maxar technologies.**


### 4.1.1 PL density and distribution

The distribution and Kernel density (KD) (search radius 1000 m) of PL centroids are shown in Figure 6 A. At the basin scale, it is possible to appreciate a high density of PL in the central sector, particularly in the Arroscia, Armetta and Tanarello valleys, where the density reaches a peak of 10 centroids / km². Moreover, Figure 6 B shows a PL detail corresponding to riverbank erosion, not filtered out because the hydrographic network has no precise high-resolution geocoding, and the derived 5m buffer did not intersect the PL. It is challenging to create an affordable geomorphological filter discriminating river erosion from a shallow landslide in a steep valley, and in many cases, the two processing overlap or are linked by a cause-effect relationship. Figure 6 C shows the correct detection of a landslide, its shape is almost accurate considering the Sentinel-2 resolution, while Figure 6 D shows a shallow landslide not detected (false negative case) because the shadow mask filters it out.

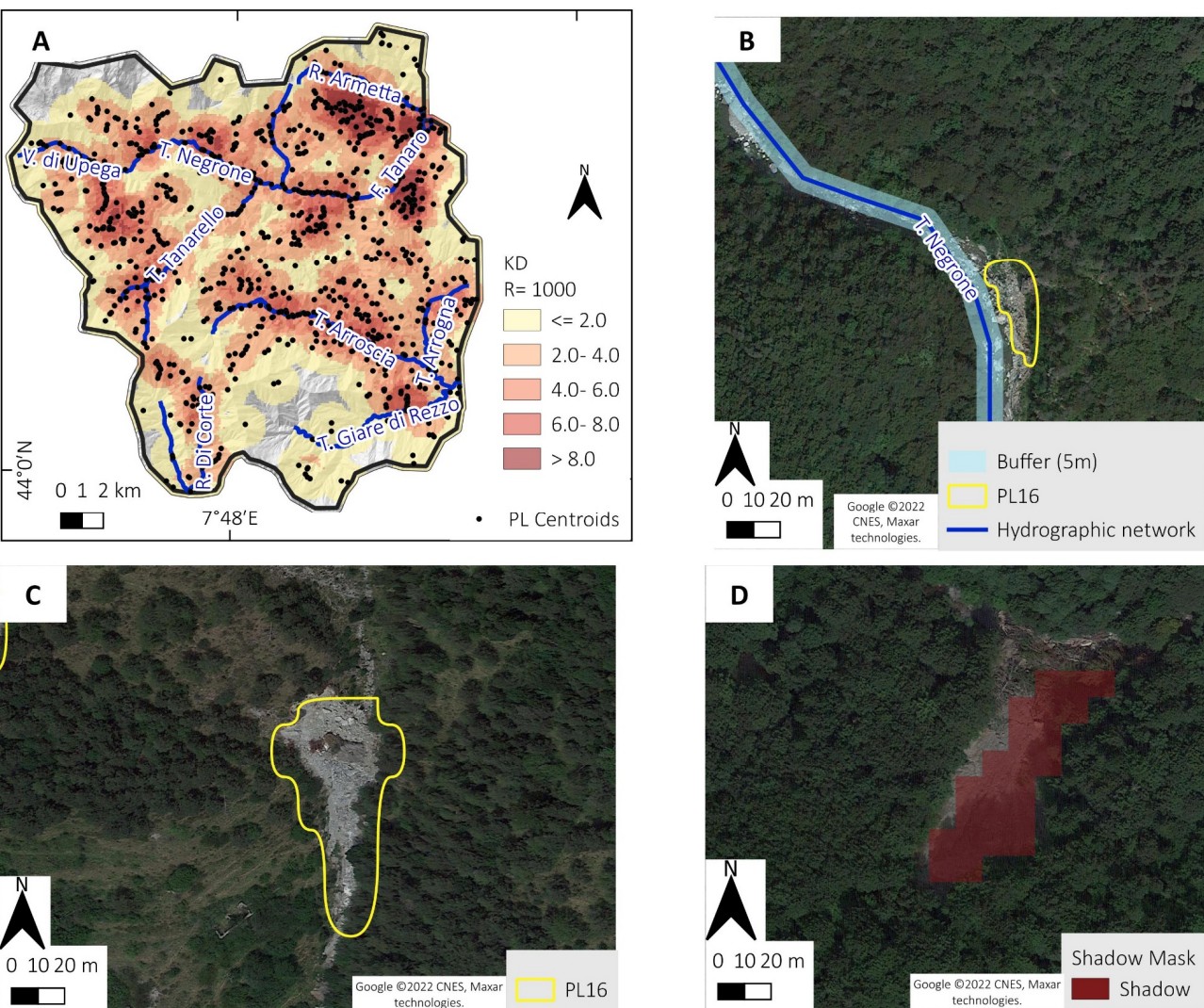

**Figure 6. A) The Kernel landslides centroids relative density (search radius = 2500). B) Detail of PL representing a river bank erosion; C) PL that correctly detects the shallow landslide, D) Shallow landslides not detected by PL because filtered with a shadow mask. Maps data: Google ©2022 CNES, Maxar technologies. The shaded relief of map A is based on 5-m DTMs of ARPA Piemonte and regione Liguria.**

### 4.1.2 PL and ML intersection results

The manual mapping of landslides made on post-event Google Earth and high-resolution satellite images was compared with pre-event aerial photos of 2012 and allowed the detection of 1098 ML (average density 3 ML/km²). The intersection of PL and ML datasets produced about 2620 cases.

Figure 7 A shows, over a sample area, some examples of the intersection between PL and ML inventories. In Figure 7 B, the intersections are classified into the five types of combinations defined in **Errore. L'origine riferimento non è stata**
**trovata.** In some cases, the PL/ML overlapping (TP) is almost complete (intersection 1) in Figure 7 A and B, while in other (intersection 2), TP cases represent a small portion of the intersection. For the whole Tanarello and Arroscia valleys case study, we also reported the intersection cases pie charts by polygons count (Figure 7 C) and total area (Figure 7 D). It is possible to note that the main difference from count to area statistics (25 % to 13 % ) is for the FN case because it corresponds to many ML with a small area. In contrast, FP shows a slight increase because the size of PL is always more
than 100m$^2$ (i.e. the Sentinel-2 spatial resolution).

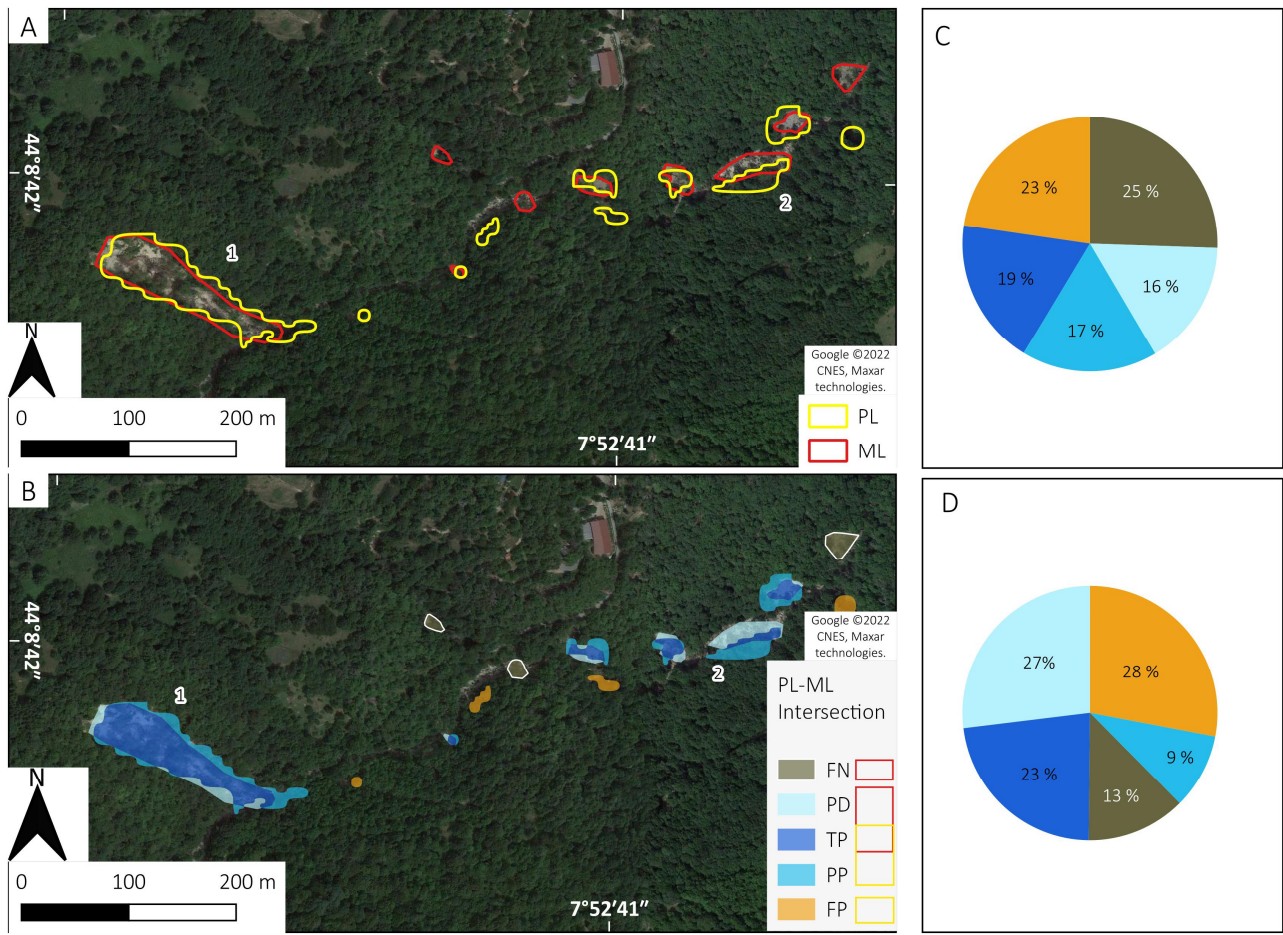

**Figure 7. Tanarello and Arroscia valleys case study. A) PL-ML inventories intersections over a sample area. B) PL-ML intersection classified by type. Pie charts of the intersection type distribution by number (C) and area (D). Maps data: Google ©2022 CNES, Maxar technologies.**


### 4.1.3 PL validation statistics.

The detailed validation results for PL made with R (see section 3.3) are shown in Figure 8. The chart of Figure 8 A shows the number distribution of TP, PP and FP cases. The TP represents about 32 % of intersected PL summed with the PP, representing 29 % of intersections, showing that the methodology correctly detects a shallow landslide in 60 % of cases. The area frequency distribution chart (Figure 8 B) indicates that the PP cases have smaller areas (median 165 $m^2$) than TP (230 $m^2$) cases, this means that the TP increase up to 38 % of the case considering the area sum instead of polygons count. Several FP cases correspond to fluvial processing like bank erosion that filters could not remove. Some other regions correspond to artificial forest cuts that occurred from 2016 to 2018. In addition, the only complete summer cloud-free pair of images is August 2016 vs August 2018, and this extended period increased the probability of detecting land-use change not related to landslides.

The $NDVI_{var}$ and the slope are the main parameters used to detect landslides, therefore, their distribution inside the intersected PL-ML inventories helped to understand the efficiency of the semi-automatic approach. Figure 8 C shows the median $NDVI_{var}$ for each class of ML-PL intersections. The TP cases show $NDVI_{var}$ values below the -0.16 threshold, probably because the shallow landslides strongly impact vegetation compared to other land-use changes. Figure 8 D shows the median slope distribution, in this case, there are few differences among the classes because the slope filter (17°) removed most FP related to the slope gradient. This means that most FP cases are caused by a land-use change or riverbank erosion with the same slope gradient of shallow landslides.

The FN and PD intersection cases are discussed in section 4.3.1, because they are not coming from PL and are not related to parameters used for PL methodology but to the relation of landslide size with the spatial resolution of the satellite.

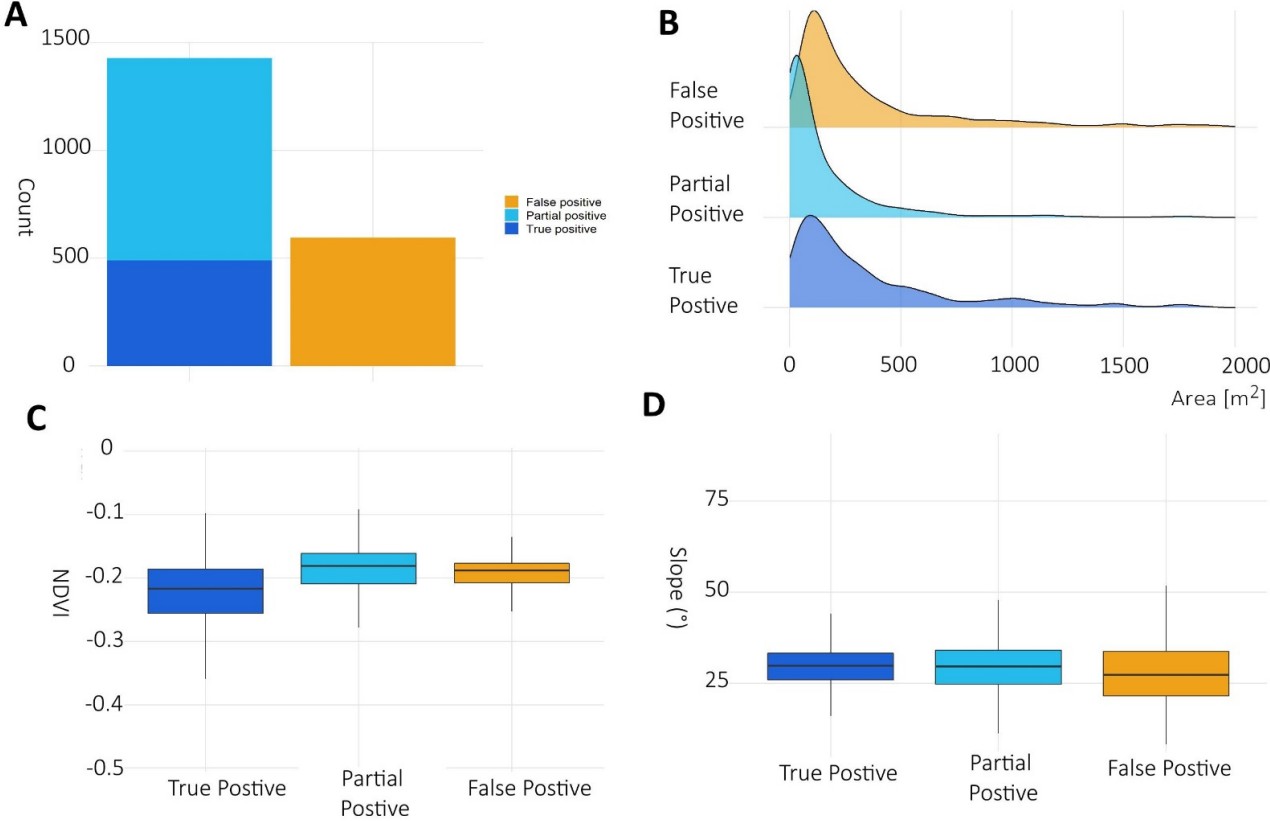

Figure 8. PL validation statistics for the Tanarello and Arroscia valleys case study: A) Bar plot showing the number of polygons of PL classified as TP, PP and FP area; B) Area frequency distribution for the polygons classified as TP, PP, and FP area C) Box plot chart of NDVI$_{var}$ distribution for TP, PP and FP classes ; D) Box plot chart of median slope distribution for each class of ML-PL intersection.

## 4.2 Gavi area case study.

For the Gavi area case study, we used the Sentinel-2 2019-06-20 (pre-event) and 2020-06-26 (post-event) images. The semi-automatic mapping allowed us to obtain about 1077 inside the training area, while about 3100 in the whole study area.

Figure 9 shows an example of the steps and results of our methodology over a sample area of the Gavi 2019 case study. From the comparison of pre- (Figure 9 A), and post-event images (Figure 9 B), we obtained the NDVI$_{var}$ (Figure 9 C), which was used to extrapolate the PL. Figure 8 D compares the ML draw on high-resolution Google Earth satellite images and the PL. A detailed 3-D view from Google Earth Pro of pre- and post-event is shown in Figure 9 E and F.

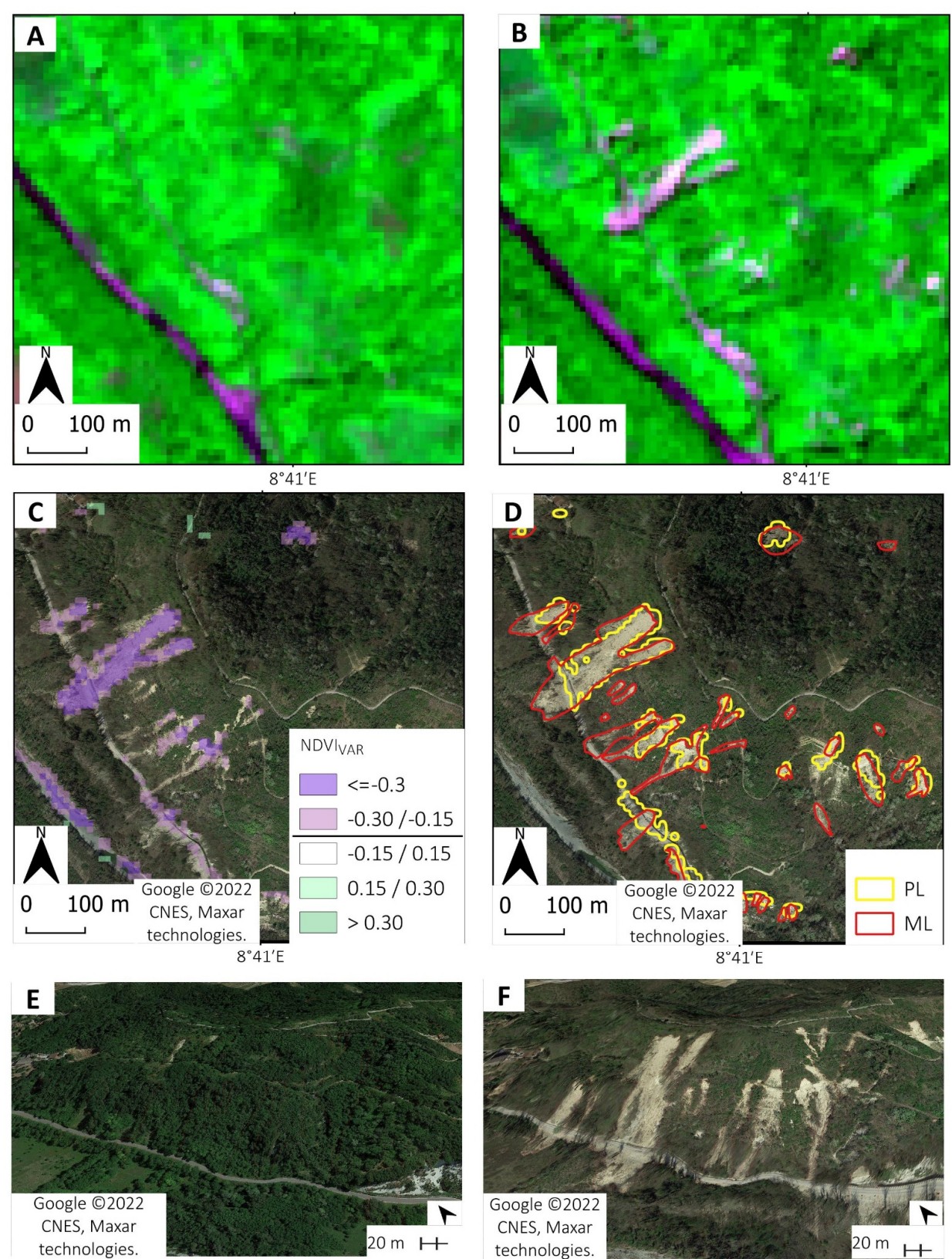

**Figure 9.** Example of shallow landslides detection methodology for Gavi 2019 case: A) Sentinel-2 pre-event images (RED-NIR-BLUE) acquired on 2019-06-20; B) A) Sentinel-2 pre-event images (RED-NIR-BLUE) acquired on 2020-06-26; C) NDVI$_{var}$ map overlapped to Google-Earth post-event image; D) Comparison of ML and PL inventories overlapped to Google-Earth post-event image. Google Earth 3D view of pre- (E) and post- (F) events of the area affected by landslides. Maps data: Google ©2022 CNES, Maxar technologies.

**4.2.1 PL density and distribution**

Figure 10 shows the Kernel centroids density (based on a search radius of 1000 m) and distributions for PL inventory (Figure 10 A). About 3000 PLs were identified using the parameters and the filters described in the methodology section, 1100 of them are inside the training area. It is possible to observe that the training and validation area show the highest PL density. Other areas of high PL density are located North of Ovada town and South of Acqui Terme. Here, manual mapping on HR images is proposed for further manual mapping. It is possible to analyze the different cases of PL methodology results in more detail. Figure 10 B shows the correct detection of a landslide, and its shape is almost accurate considering the Sentinel-2 resolution. Figure 10 C shows a PL that partly detected shallow landslides: in many cases, the trigger points are detected because of their abrupt changes, while the bottom part with more shallow sediment flow is not detected. It is also possible to appreciate two small shallow landslides not caught on Sentinel-2 images. Figure 10 D shows a false positive case where PL most probably detected a change in vegetation activity within a vineyard.

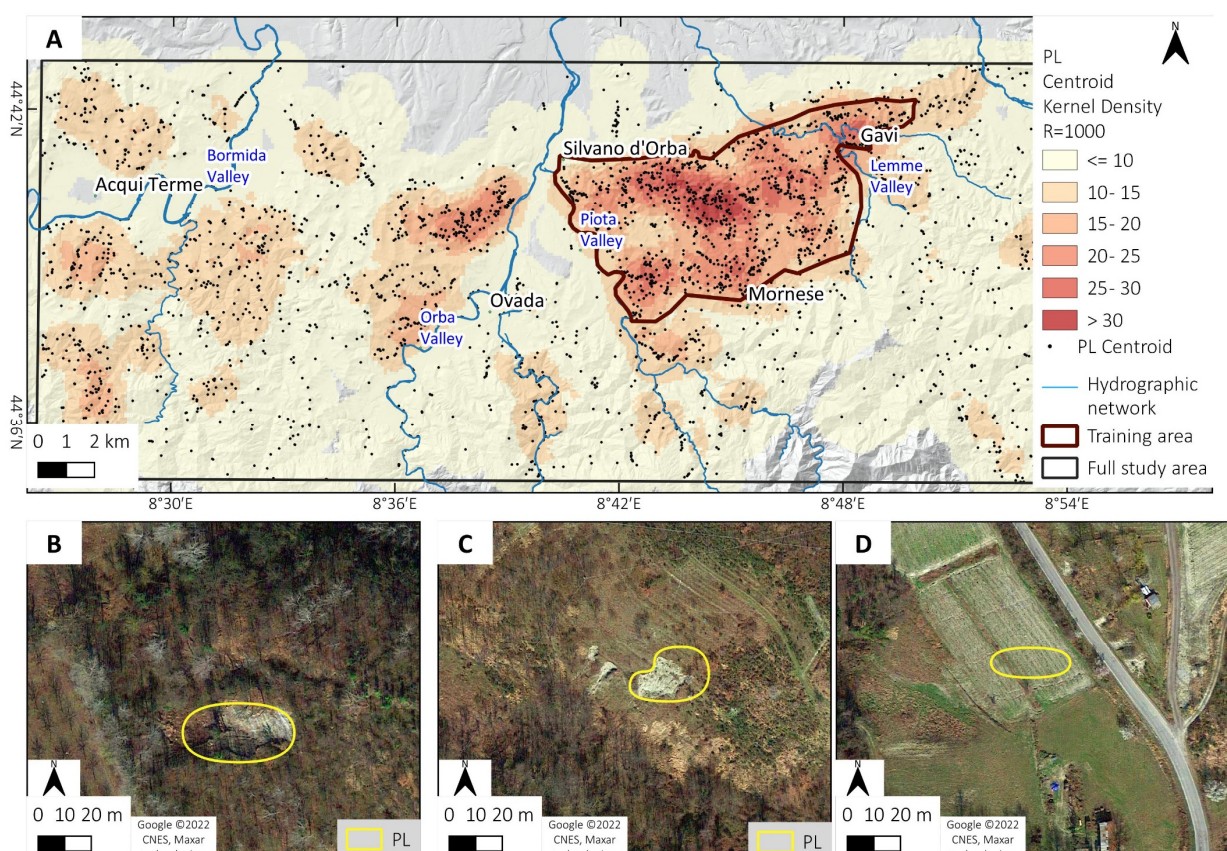

**Figure 10. October 2019 event. A) PL centroids relative Kernel Density (R=1000) over the whole AOI. Some examples of PL: B) PL that completely overlaps a shallow landslide; C) PL that partly overlaps a shallow landslide; D) PL that corresponds to a false-positive case. Maps data: Google ©2022 CNES, Maxar technologies. The shaded relief of map A is based on the 5-m DTM of ARPA Piemonte.**

**4.2.2 PL and ML intersection results**
Also, for the Gavi area case study, we made similar statistics on the inventories intersection used for the Tanarello and Arroscia valleys case study. The manual mapping of landslides, made on post-event Google Earth, and high-resolution satellite images compared with pre-event aerial photos of 2018, resulted in 1178 ML (average density 23 ML/ km$^2$). The intersection of PL and ML datasets produced about 2982 cases.

Figure 11 A shows, over a sample area, some examples of the intersection of PL and ML inventories. In Figure 11 B, the intersections are classified into the five types of combinations (TP, FP, FN, PD and PP) defined in Figure 4. In some cases, the PL/ML overlapping (TP) is almost complete (case 1) or partial (case 2). In other cases (3) the PL allowed to detect only a tiny portion of the intersection, and most of the shallow landslide area was classified as PD. For the whole 2019 study area, we also represented the intersection cases pie charts by polygons count (Figure 11 C) and area (Figure 7 D). It is possible to note that in contrast to the 2016 case, there is little difference between the count by number or total areas. The different distribution of ML size (see Figure 13 B) could explain this.

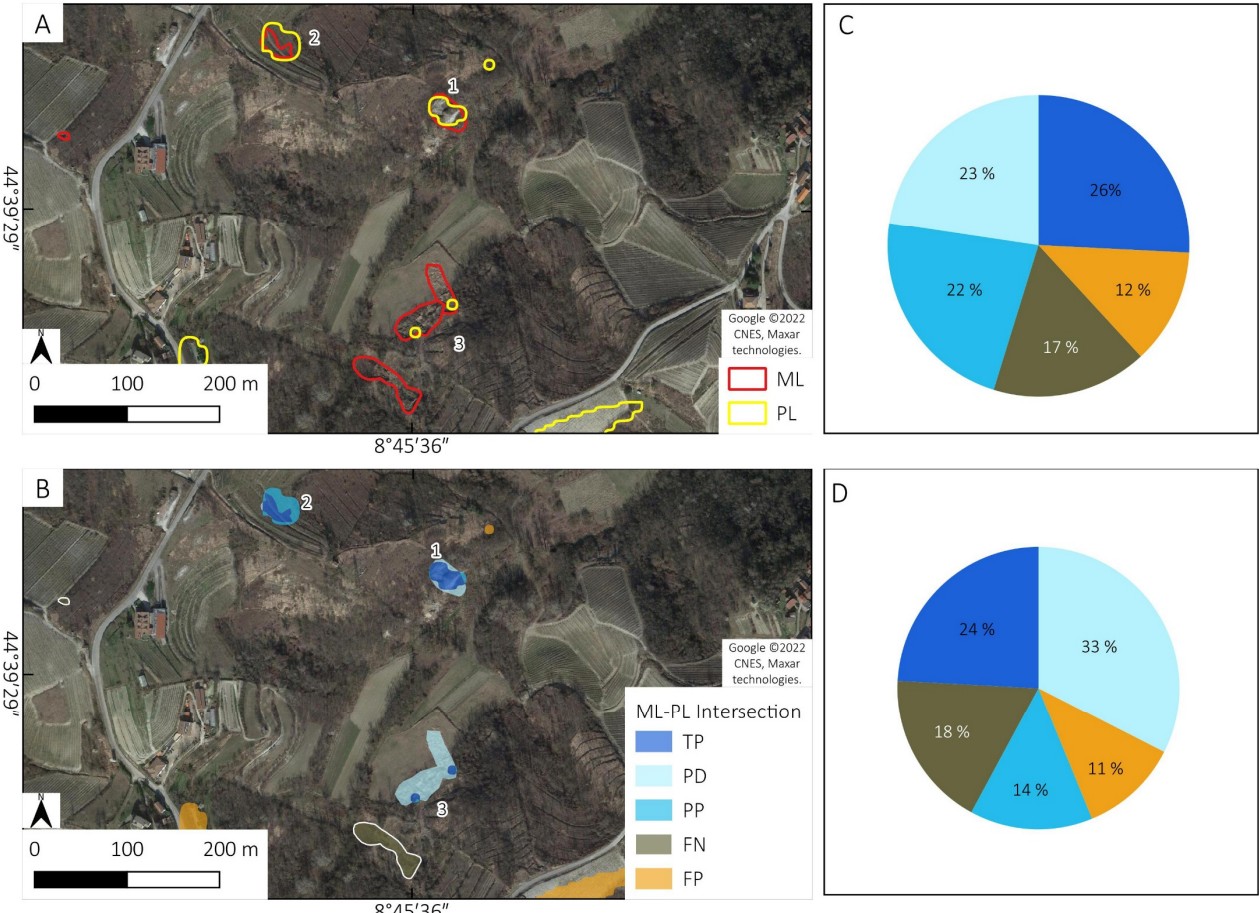

**Figure 11. 2019 Gavi case study. A) PL-ML inventories intersections over a sample area. B) PL-ML intersection classified by type. Pie charts of the intersection type distribution by number (C) and area (D). Maps data: Google ©2022 CNES, Maxar technologies.**

### 4.2.3 PL validation statistics.

The detailed validation statistics results for PL made with R (see section 3.3) are shown in Figure 12. The chart of Figure 12 A shows the number distribution of TP, PP and FP cases. The TP represents about 43 % of intersected PL summed with the PP, representing 37 % of cases, showing that the methodology correctly detects a shallow landslide in 80 % of cases. The results of semi-automatic mapping for Gavi AOI are better than Arrocia-Tanarello AOI. The better performances could be explained by a better pair of pre- post- images and the stronger magnitude of the 2019 event. The area frequency distribution chart (Figure 12 B) shows that the partial positive cases have smaller areas (median 165 m$^2$)

than TP (230 m$^2$), this implies that the percentage of TP rise from 43% to 49 % of PL considering the area sum instead of polygons count. FP cases generally correspond to changes in agriculture activity in the vineyard that occurred in 2019-2020. As previously stated, NDVI$_{var}$ and slope were analysed in terms of intersection class and provided the same insight concerning the influence of vegetation health and acclivity.

As mentioned in the Tanarello and Arroscia case study (section 4.1.3), The FN and PD intersection cases are discussed in section 4.3.1

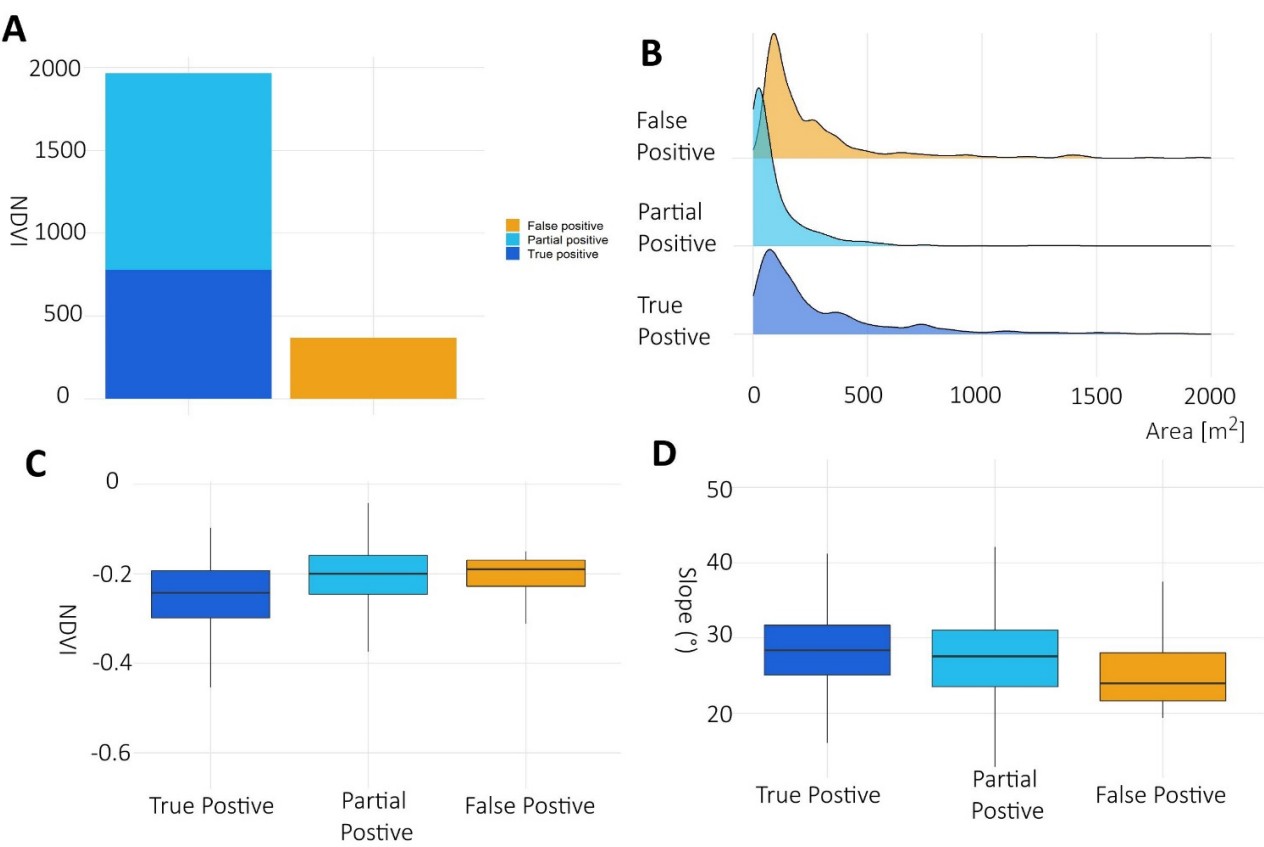

**Figure 12. PL validation statistics for the 2019 Gavi study area: A) Bar plot showing the number of polygons of PL classified as TP, PP and FP area; B) Area frequency distribution for the polygons classified as TP, PP, and FP area C) Box plot chart of NDVIvar distribution for TP, PP and FP classes ; D) Box plot chart of median slope distribution for each class of ML-PL intersection.**

**4.3. Factor that influences shallow landslides detection and false positives.**

In this chapter, we discuss the main factors that, according to our results, influence the efficiency of the proposed PL methodology. As already treated in the literature, the landslide size (Bellugi et al., 2021; Fiorucci et al., 2018), land use (Mondini et al., 2011) and the temporal interval (Lindsay et al., 2022; Scheip and Wegmann, 2021) used in the pre- post-event calculation seem the most interesting to discuss on the base of our results.

### 4.3.1 Shallow landslide size, density and distributions.

The detection capacity is mainly related to shallow landslides size and spatial resolution of Sentinel-2.

Figure 13 A and B show the histogram distribution of ML size for the Tanarello and Arroscia valleys and Gavi study areas, respectively. ML size distribution agrees with the classical power-law distribution of shallow landslides (Bellugi et al., 2021; Guzzetti et al., 2002) for both cases study. This means that the shallow landslides smaller than 500 m$^2$ are about 60% of all ML inventories, but they represent only 20 % of the total area affected. The histograms also show that for the Tanarello and Arroscia valleys (Figure 13 A), the MLs are generally smaller than in the Gavi area. The parts of the bars black coloured show the ML intersected by PL (TP+PD intersection cases). It is possible to note that the small landslides (100 m$^2$) are underestimated by PL, while for large landslides, PL almost fit the ML. These results agree with the area frequency distribution for TP and PD cases shown in Figure 8 B and Figure 12 B.

Figure 13 C and D show the DR calculated with equation 2 for the Tanarello and Arroscia valleys and Gavi study areas. The DR increases with the size of the landslides, and it is strictly related to the pixel size of Sentinel-2. For the landslides smaller than 100 m$^2$ (Sentinel-2 spatial resolution) the DR range from 10% of Tanarello and Arroscia Valley (Figure 13 C) to 15 % in the Gavi area (Figure 13 D). It is clear that for these landslides, the Sentinel-2 platform is not the optimal choice (Fiorucci et al., 2018), and manual mapping on high-resolution images is better than semi-automatic mapping. On the other hand, for the ML in the class area (500 – 1000, i.e., 5 – 10 pixels of a Sentinel-2 image), the DR range from 65 % of Tanarello and Arroscia Valley to 80 % of the Gavi area.

The overall DR considering polygons count is 39 % for the Tanarello and Arroscia Valley and 58 % for the Gavi case study, while considering the area sum, the DR is 60 % and 75%, respectively. The Gavi case study's better performance is likely related to the size distribution of shallow landslides and the better pair of images used to create PL.

The performance metrics Precision (P) P= TP/(TP+FP), Recall R= TP/(TP+FN) and F1-Score F1= 2TP / (2TP + FP + FN) without considering the intersection cases are reported in Table 3. The overall performance of our methodology is comparable with other studies, especially those using middle-resolution images (Ghorbanzadeh et al., 2021; Mondini et al., 2011; Handwerger et al., 2022). A recent study (Ganerød et al., 2023) using Sentinel-2 shows different performances depending on the study area in Norway; the best is from the deep-learning approach with a U-net architecture. Using high-resolution Planet images (Bhuyan et al., 2023) achieves high performance over several study areas. However, we assume that the different study/training/validation settings, the image used, and the event type made this comparison relative.

**Table 3. Performance metrics of this methodology.**

| AOI | P | R | F1 |
|---|---|---|---|
| Arroscia / Tanarello 2016 | 45 % | 64 % | 0.529 |
| Gavi 2019 | 69 % | 57 % | 0.623 |

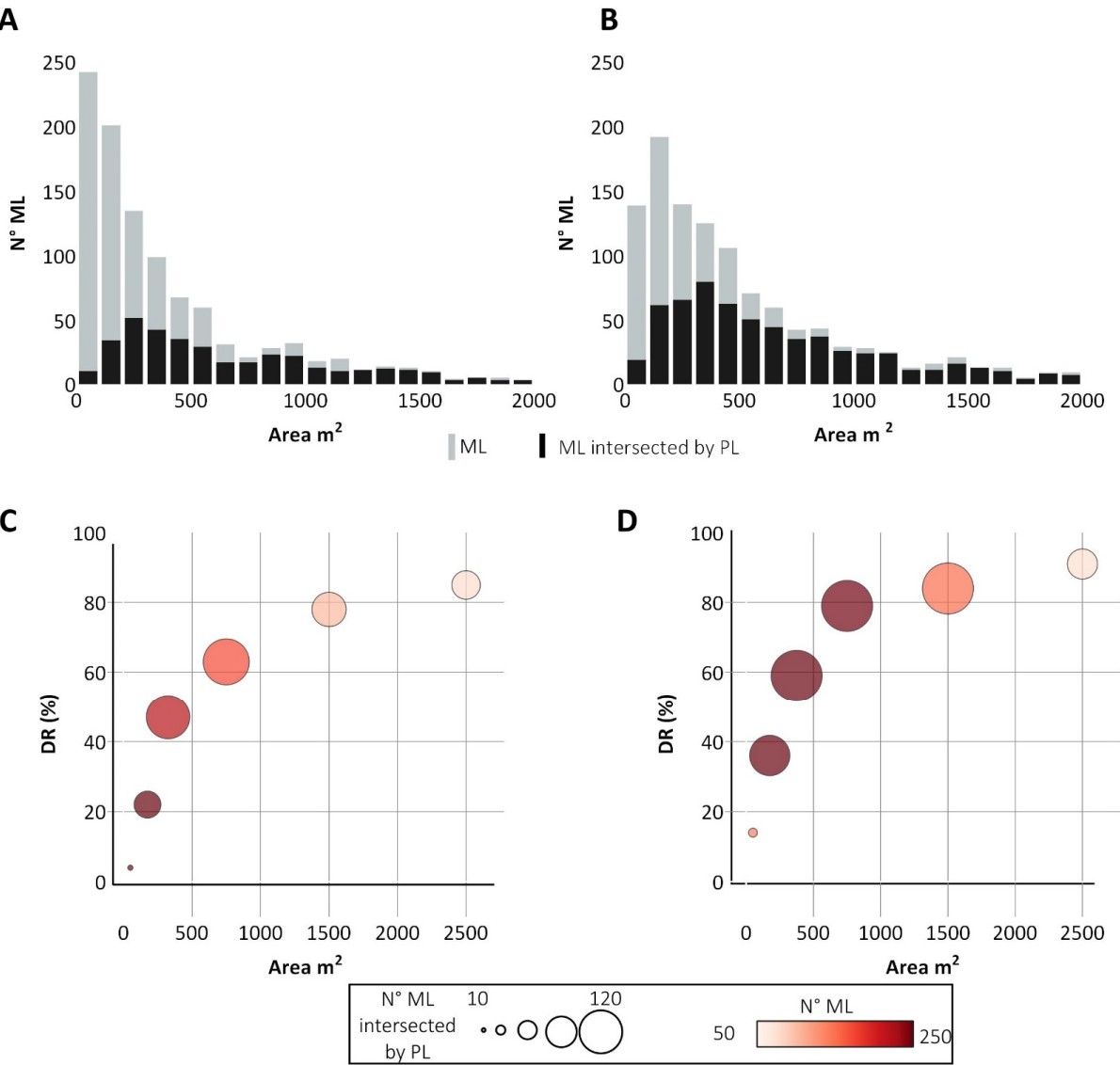

**Figure 13 Histogram of ML area distribution for Tanarello and Arroscia valleys (A) and Gavi (B) study areas, the grey bars represent the entire ML inventory while the black one represents the ML intersected by a PL. Detection rate (DR) for each area class for Tanarello and Arroscia valleys (C) and Gavi area (D) case study. The size of the circle is correlated with the number of detected ML; the colour scale represents the total number of ML for each class.**

Both in Tanarello and Arroscia valleys and Gavi areas, the results of spatial correlation of ML and PL density show a good agreement (Figure 14), a further confirmation of the effectiveness of the PL methodology. Specifically, in the case of 2019, both ML and PL inventories, Figure 14 A and B, show a higher density in the northern sector of the study area and a lower density in the central region. Figure 14 C shows the scatterplot between the PL and ML density, the correlation coefficient (Benesty et al., 2009) has a value of 0.85 with an $R^2 = 0.73$ using a grid of 1000 m. Moreover, the average ML density is about 23 ML/km² higher than the 2016 event (3.1 ML/km²). The difference is probably related to the more intense rainfall of October 2019 (up to 500 mm in 24 h) compared to the 2016 case (700 mm in five days). +

We also manually checked on Google Earth and noticed the same performance in a small validation area outside the Gavi training area.

In the case of 2016, there are some more discrepancies. It is possible to appreciate a higher density of PL in the NE sector (Figure 14 D). At the same time, ML shows a density peak in the Arroscia valley (Figure 14 E). Both datasets show that centroids' high density is located in the central sector, particularly in the Arroscia, Armetta and Tanarello valleys, where the density reaches a peak of 10 centroids/km². At the basin scale, PL/ML regressions show a correlation coefficient of 0.68 with an $R^2$= 0.47 using a grid of 2500 m (Figure 14 F).

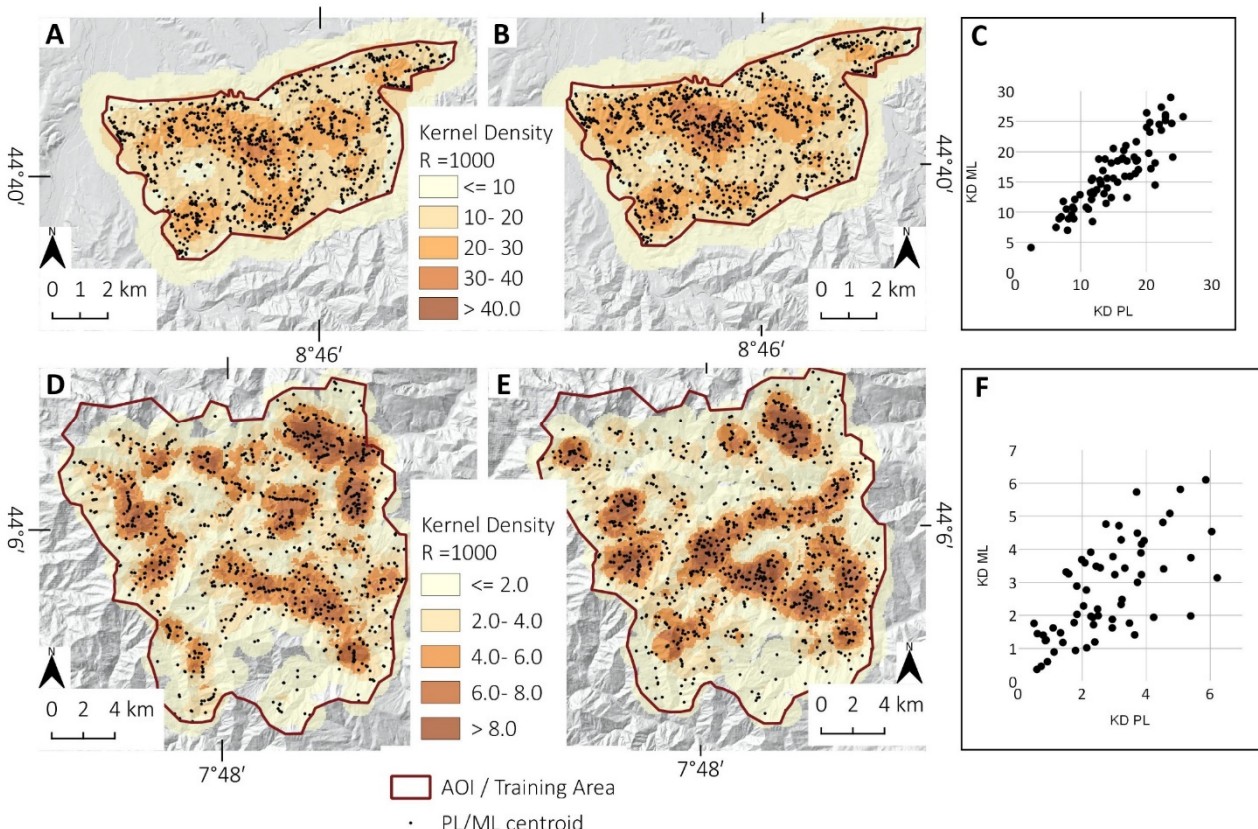

**Figure 14. Heatmap Kernel Density for: A) Gavi PL inventory; B) Gavi ML inventory; D) Tanarello and Arroscia valleys PL inventory; E) Tanarello and Arroscia valleys ML inventory. Scatter plot of ML/PL Kernel density comparison for: C) Gavi training area; F) Tanarello and Arroscia valleys study area. Shaded relief of maps A, B, D and E are based on 5-m DTMs of ARPA Piemonte and regione Liguria.**

### 4.3.2 The effect of land use on PL methodology efficiency.

We deeply investigate the role of land use in landslide detection. The semi-automatic detection performed well for naturally vegetated slopes, while the detection capability decreased for cultivated landscapes.

For instance, the statistics on intersection cases for the four main land-use classes (Land Cover Piemonte, 2022) made for the 2019 case study allowed us to understand the variable regarding the land-use efficiency of semi-automatic mapping.

The forest and the new forested shrub areas (Figure 15 A and B ) show a high percentage of TP cases, and about 80 % of PL corresponds to shallow landslides. Here, human disturbances are limited, and most NDVI$_{var}$ are related to landslides. The FN (15% of the area) fits small landslides not detectable with Sentinel-2 because of its spatial resolution.

The vineyard land use (Figure 15 C) shows a high FP percentage (37%). In this land use, most FP cases are changes related to vineyard management. FN are also higher (22%) compared to the forested areas. The cultivated land (Figure

15 D) shows a high percentage of FN (30%). The underestimation can be explained by smaller landslide dimensions and
570 the agricultural practice that erases the signs of landslides.

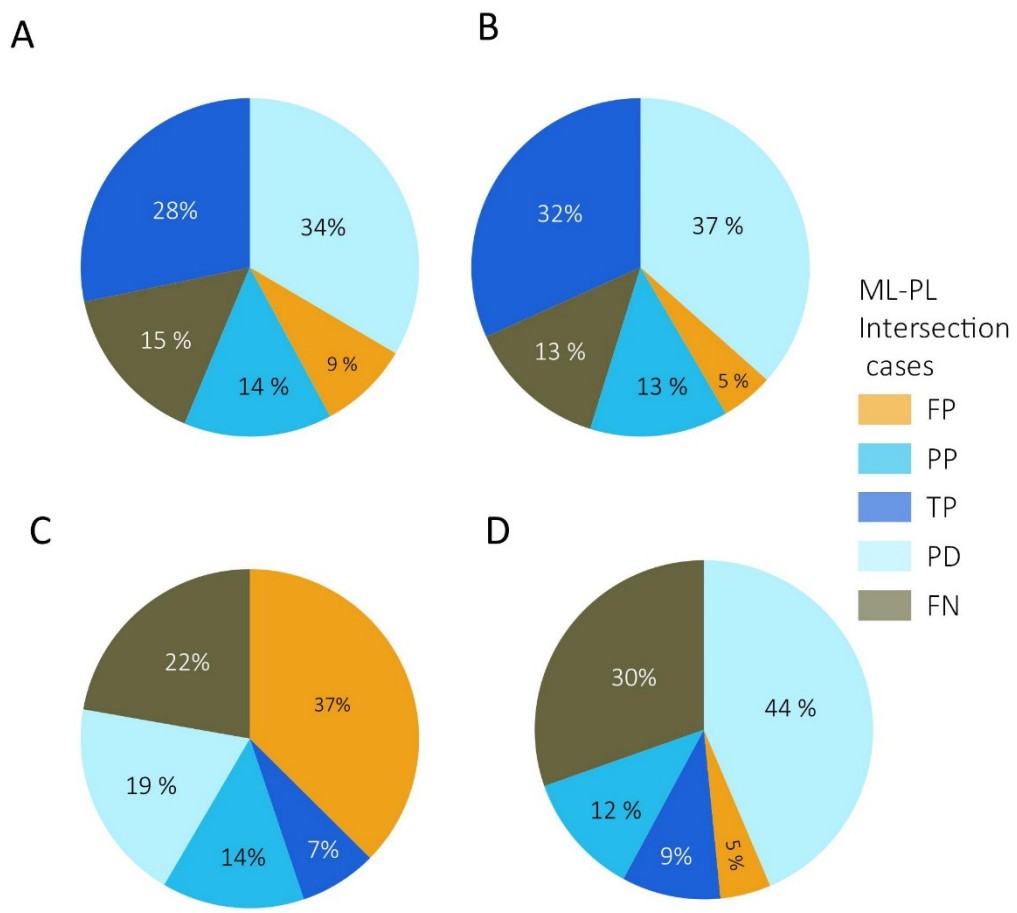

**Figure 15. Gavi 2019 case study, PL-ML intersection distribution for the main land-use classes: A) Broad-Leaf forest; B) New-forest or shrubland; C) Vineyard; D) Cultivated land**

**4.3.3 NDVI time series based on GEE**

Figure 16 A and B show the NDVI time series based on Sentinel-2 computed on GEE (see data availability sections). The NDVI time series of the area affected by shallow landslides (red line in Figure 16) is based on averaged NDVI of a subset of ML polygons with an area of at least 500 m$^2$. This time series is compared with the time series averaged over a region
not affected by landslides (black line in Figure 16).

In the case of the Arroscia-Tanarello 2016 study area (Figure 16 A), it is possible to observe that in the areas not affected by landslides, the NDVI has a relatively constant seasonal trend during 2015-2021, ranging from 0.25 in winter to 0.75 in summer. On the contrary, the areas affected by shallow landslides showed a substantial decrease of NDVI (from 0.75 to 0.4) in the post-event 2017 summer. The NDVI slowly increased during the following years, up to the value of 0.55 in
2021. If we compare the pair images (2016-08-23, 2016-08-28) used to extract the PL, we can observe an average NDVI$_{var}$ of -0.2. Unfortunately, the summer 2017 images that show the best NDVI$_{var}$, are not fully cloud-free or acquired in the same seasonal period.

In the case of the Gavi case study (Figure 16B), in those areas not affected by landslides, the NDVI has a constant seasonal trend during the period 2016-2021, ranging from 0.2 in winter to 0.8 in summer. In contrast, the NDVI of the areas affected by shallow landslides shows a substantial decrease, showing a value of 0.6 (-0.2 variation) in the 2020 summer. The NDVI value rebounded to 0.65 in 2021. The pair images (2019-06-26, 2020-06-20) used to extract the PL show an average $NDVI_{var}$ of -0.15. The vegetation seems to recover faster compared to the 2016 case.

The effects of land use on detection capabilities are difficult to solve with simple filters, a partial solution could be the NDVI time series analysis. The analysis of the single-pixel time series of NDVI, generated with the GEE QGIS plugin (GEE Timeseries Explorer), allowed us to understand better the effect of landslide size and land use in the capacity detection of Sentinel-2, the main results are resumed in Figure 17.

Figure 17 A and A' show the location and the comparison of two NDVI time series in a broadleaf forest land use: the 1-p is entirely inside a ML of approximately 1500 $m^2$, while the 2-p is an undisturbed area. The two time series show almost the same seasonal trend until the 2019 event, after this, the NDVI of 1-p shows values of 0.2 less than the NDVI of 2-p in the 2020 summer.

Figure 17 B and B' show the location and the comparison for two single-pixel time series located in a vineyard land use. In this case, the (5-p) is situated in ML with an approximate size of 500 $m^2$ not detected by PL methodology (a false negative case). Both slide (5-p) and no-slide (6-p) NDVI time series show an irregular trend, probably related to the agricultural activity inside the vineyard. The post-event time series shows lower NDVI values for the pixel inside the ML, however, the noise is too high to extract a clear trend.

Figure 17 C and C' show the location and NDVI time series of a false-positive PL in a vineyard land use (9-p). In this case, the agricultural activity inside the vineyard caused a $NDVI_{var} < -0.16$ using the pre- and post-event pair images. However, if we look at the whole time series, there is no trace of the effect of shallow landslides.

Instead of considering single pre (2019-06-26) and post (2020-06-20) event dates, the noise could be reduced if we use averaged the pre- and post-event summer (May-September) images. Multi-temporal averaged satellite data on GEE have already been used to improve landslide detection (Lindsay et al., 2022). For instance, using this approach, the FP of Figure 17 C disappears because the $NDVI_{var}$ is -0.16 for single pairs while it is -0.03 for the averaged summer periods.

The summer averaged pre- and post-event NDVI computed with GEE for the Gavi area are available in the data availability sections.

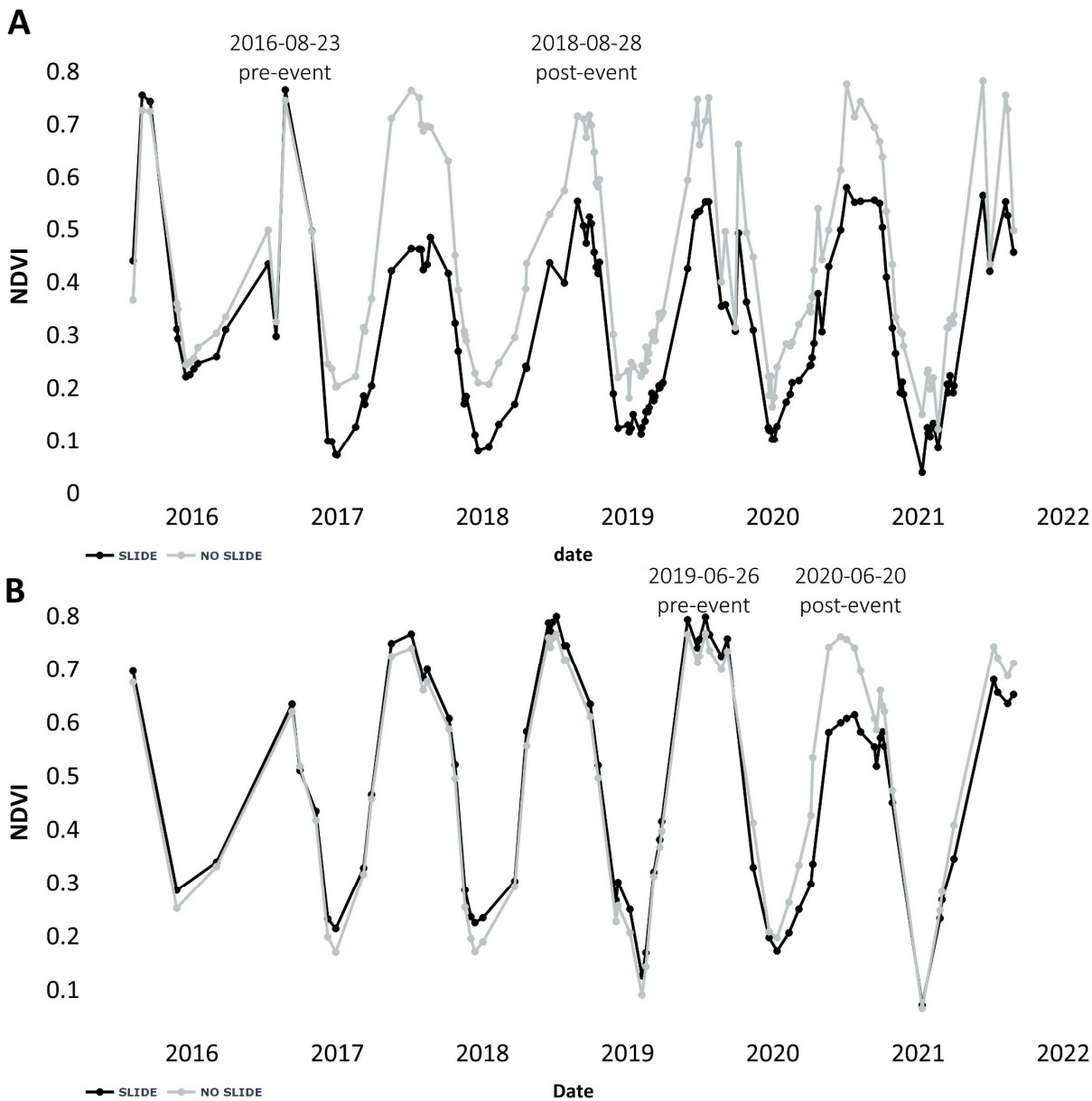

**Figure 16. NDVI Time series based on Sentinel-2 data and processed with GEE: the 'Slides' black lines are the averaged time series of the selected ML; the 'no slides' grey lines are the time series of the surrounding unaffected areas. A) 2016 Arrocia-Tanarello case study; B ) 2019 Gavi case study.**

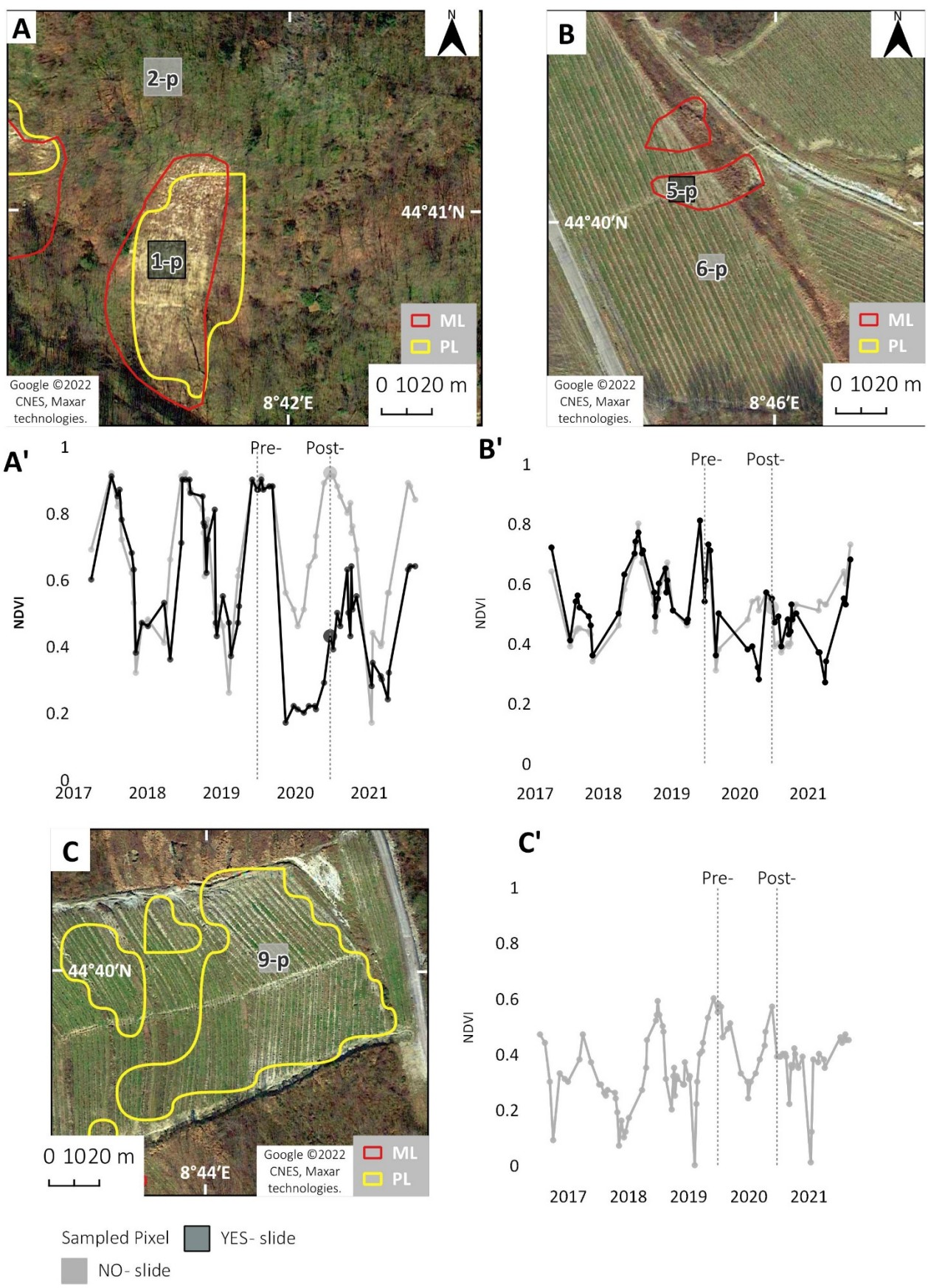

**Figure 17. NDVI Time series (2017-2021 of five sample points (Landslide / No Landslide) in the area affected by the 2019 event. A and A' comparison located in a broadleaf forest (true positive); B and B' vineyard (false negative); C and C' vineyard (false positive). The Google Earth satellite image was acquired on 2021-03-16. Maps data: Google ©2022 CNES, Maxar technologies.**

## 6. Conclusions

A low-cost semi-automatic methodology to detect shallow landslides using cost-free data from Sentinel-2 satellites, was implemented and tested in exemplary cases. The implemented user-friendly processing is aimed to be used by a wide range of final users, such as technicians or natural hazard planners with low expertise in remote sensing processing and computer science. Landslide data and GEE scripts were also available in the free storage database for the researcher and community users. We tested the methodology on two extreme rainfall events that affected NW Italy in November 2016 (Tanerello and Arroscia valleys) and October 2019 (Gavi area).

We obtain a polygons dataset representing potential landslides (PL) and their density maps with the semi-automatic mapping. The PL density maps aim to identify the most affected areas, where to focus the manual mapping on high-resolution images. The PL dataset is created on three steps approach: i) choice of the best pair of Sentinel-2 images also with the help of NDVI time series computed with Google Earth Engine (GEE); ii) creation $NDVI_{VAR}$ map from pre- and post-event images, iii) geomorphological filters based on slope and others parameters such as distance from the hydrographic network, iv) eventual re-calibration of PL parameters with landslides manually detected (ML). In the study areas of Tanarello and Arroscia Valley, and Gavi, 1077 and 2576 PL were detected, respectively. The areas with the highest PL density also match with most affected areas by rainfall events.

The PL inventory was compared with manual mapping based on high-resolution images in training and validation areas. We manually map landslides on free-cost high-resolution images (e.g., Google Earth, national or regional cartographic services). The manual mapping detected about 1100 ML over 300 km$^2$ in the Tanarello and Arroscia valleys and about 1180 ML over 55 km$^2$ in the Gavi area. The PL datasets were then compared (ML). The ML and PL inventories comparison shows a good agreement in density and distributions, especially for the Gavi case study.

According to the findings, the semi-automatic method can detect the majority (about 60 %) of shallow landslides larger than two or three times the size of Sentinel-2 ground pixels (100 m$^2$). In contrast, the PL method can identify only 20% of small landslides ( less than 100 m$^2$ in size). In agreement with the power-law, small landslides are a high number, but a small fraction of the affected area. Consequently, the false-negative represents 60 % to 40 % of the cases but only 20-25 % of the area affected. In the future, using high-resolution cost-free satellite images for semi-automatic methods could drastically increase the detection capacity. Land use is another factor that influences detection capacity: in the Gavi study area, the detection rate reached 73 % on natural vegetation like forest land use, while in cultivated areas, DR decreased up to 41 %,

The false-positive rate (FPR) (28 % for the Gavi area and 48% for Tanarello and Arroscia valley) is related to riverbank erosions or artificial change in vegetation pattern that was impossible to remove with geomorphological filters. Also, for FPR, the land use drives the performance: it ranges from 45 % of the vineyard to 20 % of the forest.

In summary, the PL inventory has strengths, such as its ability to: quickly map large areas, cost-effectiveness, and user-friendly processing. However, it also has limitations, including low resolution and a higher likelihood of false positives. Therefore, the primary purpose of the PL inventory is not to generate precise susceptibility or damage assessment maps. Instead, it serves as an initial and swift assessment map, allowing users to identify the most heavily impacted areas. These areas can be targeted for further detailed mapping or on-site field surveys. The analysis of the NDVI time series with GEE was proper to identify the best pair of images and monitor the behaviour of vegetation of affected areas. The best comparison is in the period of maximum vegetation activity (e.g., summer images for middle latitude). The time series showed a progressive vegetation recovery throughout the years, decreasing the detection capacity of this approach. The

time-series analysis also suggested that an averaged NDVI of summer periods could reduce the effect of agricultural changes on vegetation and limit the false positive. The evidence indicated that multi-temporal analysis needs to be developed to improve semi-automatic mapping efficiency, which could be the key to future studies.

**Data availability**

ML and PL datasets used in this study are available in KML format on the Zenodo platform (Notti et al., 2022)

The Google Earth Engine scripts used for this paper are:

- NDVI Time series of some sampled areas to select the best pair of images for the PL creation (Tanarello and Arroscia Valleys and GAVI AOIs (Fig. 15 of the manuscript) https://code.earthengine.google.com/8f4d08db4d34099bb1a4cd1fd15bf05b?noload=true

- Sampled NDVI Time series from different intersection cases for Tanarello and Arroscia Valleys study area https://code.earthengine.google.com/47e5883e0fe7beae589df4c42421dd3b?noload=true

- Sampled NDVI Time series from some test areas over the Gavi study area https://code.earthengine.google.com/e97fd46102cca3ee25b441a8ebff8400?noload=true

- Multi-temporal averaged $NDVI_{var}$ for the whole Gavi study area
https://code.earthengine.google.com/454fd113a211fb8b745f5aa658da234c?noload=true

The Sentinel-2 images were downloaded from Copernicus ESA open-access hub website: https://scihub.copernicus.eu/

The High-resolution images were uploaded in QGIS as WMS layers from the following link:

- Google Earth Satellite layers, XYZ tiles for QGIS: http://mt0.google.com/vt/lyrs=s&hl=en&x={x}&y={y}&z={z}

- The 2012 Orthophots that cover both study areas are available on the national cartographic service (PCN) WMS: http://www.pcn.minambiente.it/mattm/servizio-wms/ Accessed: 2022-04-20

- The Piemonte region 2018 orthophoto is available on the following WMS: https://opengis.csi.it/mp/regp_agea_2018?service=WMS&request=GetCapabilities&version=1.3.0, The Regione Liguria 2016 Orthophoto are available on
http://www.cartografiarl.regione.liguria.it/mapserver/4.10/mapserv.exe?MAP=E:/Progetti/mapfiles/repertorioc artografico/ORTOFOTO/1828.map&VERSION=1.3 Accessed: 2022-04-20

DTM data can be downloaded from Regione Piemonte and Liguria geo-portal:

- https://www.geoportale.piemonte.it/geonetwork/srv/api/records/r_piemon:224de2ac-023e-441c-9ae0-ea493b217a8e Accessed: 2022-06-30

- https://srvcarto.regione.liguria.it/geoviewer2/pages/apps/geoportale/index.html?id=2056 Italian) Accessed: 2022-06-30

Google Earth Pro software was also used to map and check inventories at high resolution.

Other data, such as land use, NDVI elaborations and meteorological data, can be obtained from the first/corresponding author upon reasonable request.

**Acknowledgements**

This work was supported by Major International (Regional) Joint Research Project of NSFC (No. 42020104006)

**Author contribution.**

Planning and conceptualization were done by DN. Responsible for data curation were DN, MC, and DGo. The mapping was done by DN and MC. The statistical analyses were performed by DGo. The original manuscript was written by DN.

MC, DGo and DGi were involved in reviewing and editing of the manuscript. Supervision and funding acquisition were provided by DGi.

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
