# Peer review of "Semi-automatic mapping of shallow landslides using free Sentinel-2 images and Google Earth Engine."

_Natural Hazards and Earth System Sciences, 2022_

## Author Comment (AC1)

NHESS comments

**Rew1**

This manuscript presents a practical procedure for using free data of Sentinel-2 and Google Earth Engine to create an inventory of shallow landslides. The procedure was applied to two landslide areas and the results were compared with the manual landslides. Overall, the methodology is well presented and the data analysis is performed in detail. The methodology shows the potential to create inventories of landslides that are fundamental to landslide research. I have some minor comments for the authors to consider:

**We thank the reviewer for her/his positive comments and the very useful revision and constructive suggestions. We reply here to the minor comments suggested by the reviewer. Meanwhile, we are working on an improved manuscript and figures version. We will upload the revised when the revision process is completed with other reviewers reports.**

(1) Figs. 1 and 2 included the rock formation of the studied sites. What is the role of geological information in the creation of landslide inventories?

**R: Thank you for the observation. The geological information is not used in the creation of landslide inventories (PL and ML), however, they are essential ancillary information added to the database for statistical analysis of the landslide density distribution or future studies on landslide susceptibility. In addition, the geological setting is important to understand the context in which the events occurred.**

(2) Line 190, Eq.1, could the authors elaborate how the eq. 1 is applied to determine the NDVI value of the studied area? What is the 'pixel' size ( the calculation area) for each of single NDVI?

**R: The equation 1 (the NDVI calculation) is made using the rasters of a single band of Sentinel-2 (Red-band 4, NIR-band 8) and calculated with the raster calculator of QGIS. The NDVIvar is a simple difference of pre-post event NDVI also calculated with the QGIS raster calculator tool. The same equation can be used on Google Earth Engine or any other software that manages raster data. The pixel size corresponds to the spatial resolution of the bands used (10 m for red and NIR bands of Sentinel-2) consequently, the NDVI raster also has 10 m of pixel size. In the new version of the manuscript, we add a more detailed explanation of this point**

(3) The authors should detail how a polygon of PLs is formed?

**R3: The rasters of $NDVI_{Ivar}$ and slope are converted in a Boolean raster (0-1) using the thresholds in a rater calculator of QGIS (e.g. B = NDVIvar<-0.16 and slope>15). In the obtained boolean raster, the values 1 correspond to the potential landslides (PL). Then on QGIS, we converted the values 1 of raster into vectors (polygon). Then, the median slope (always with QGIS) is calculated for each PL polygon and filtered with a certain threshold. The polygons are also smoothed to obtain a shape more geomorphological based. As the creation of PL from raster could be made with several GIS software, describing the detailed procedure used with QGIS on the manuscript could be too specific.**

(4) Reference to Eq. 2 and Eq. 3 should be provided. The definitions of Eq. 2 and Eq. 3 seem to be different from other studies.

**R4: We used these equations specifically for this study, there are no specific references available. However, similar equations can be found in several works on landslide detection (Catani et al., 2021, Prakash et al., 2021, Volpe et al., 2022, Meena et al., 2022), and we add them to the manuscript. In this work, we considered complex intersection cases. In addition to the simple true positive, false negative, and false positive cases used in other studies, we have the partial intersection. It was necessary to take into account also the partial intersection because of the different spatial resolutions of the images: PL are based on Sentinel-2 images (10 m spatial resolution). At the same time, ML are mapped on high-resolution images with 0.5 m of spatial resolution. In many cases, PL and ML polygons shapes did not fully match because they were based on two different images.**

(5) Figs. 4 and 8 C, could the authors remove the PL data to clearly show the color map of NDVI?

**R5: Thank you for the helpful suggestion. In the revised version, we improved these figures.**

(6) Fig. 13, there is no 'C' in the caption. Also, comment on how C is created. From A and B, it seems that the kernel density points/locations are slightly different.

**R6: Thank you for the comment, the caption has some errors referring to the sub-figures. In the new version of the manuscript, we will provide to add the description of 'C' in the caption. Yes, the kernel points' density and distribution are different because they are derived from PL and ML datasets, respectively. Figures C (Gavi area) and F (Tanarello area) show the scatter plots of PL (x-axis) and ML (y-axis) median density calculated over regular grids (1x1 km for 'C', and 2.5x2.5 km for 'F') on QGIS. From these data, we also calculated the coefficient of correlation.**

(7) The methodology still needs to be improved to capture the landslide inventory accurately. Could the authors comment on the limitations and the area for improvement?

**R7: Thank you for the comments. The main limitation is the spatial resolution of the satellite: basically, the landslides smaller than satellite spatial resolution largely underestimated. Using commercial satellites with high spatial resolution would improve the number and the shape accuracy of landslides detected, which implies a higher cost of mapping. Other improvements can be related to the filters used to select the PL on slope and NDVI index. Also, as you suggest in the following comment, the use of machine learning could be an improvement**

(8) Instead of having operators change the mapping parameters and visually compare the ML and PL, are there any qualitative ways (e.g. machine learning, optimization methods...)to get the most appropriate parameters to have the highest degree of match between ML and PL?

**R8: Thank you for the observation. The use of machine learning or multivariate analysis would undoubtedly be an improvement for the detection of shallow landslides. However, machine learning development that identifies landslides needs a dedicated investigation and training on a considerable amount of data. The methodology presented in this was focused on a fast and user-friendly approach that allows not skilled people in remote**

sensing or coding to create a map of potential landslides. The proposed methodology could be an interesting topic for future study, in this sense, we also published the dataset of PL and ML (https://zenodo.org/record/6617194) to encourage the community to improve our methodology.

---

## Author Comment (AC2)

The study of Notti et al. deals with the semi-automatic recognition of shallow landslides for the compilation of event-based inventories over two study areas in NW Italy. In general, complete landslide inventories are very important products for the study of susceptibility, hazard and risk. Therefore, I retain this study a very interesting contribution for NHESS. Before publication, I would like the authors to answer a couple of general comments and a series of detailed comments, all listed below.

**The authors would thank revisor#2 for his good evaluation and constructive comments that helped improve the manuscript. The replies point-by-point are reported here following.**

**The revised version of the manuscript also contains the track changes.**

**General comments**

- In my understanding, the manual delineation of landslides for the calibration and in this case the validation of the automatic procedure is the most time-requiring activity related to the proposed procedure. For repeatability in other areas/regions, would it be possible to structure the study so to give an indication of how large the training areas should be, compared to the total investigation domain, to allow for quality (high performance) results?

**R: Thank you for your helpful observation. Our work aimed to obtain a semi-automatic map of shallow landslides based on free-cost images and devoted to an immediate post-event application. To obtain a robust procedure, a validation based on the comparison with the manually mapped landslides has been planned, too. In general, regarding the size of the training area, there is no fixed proportion threshold between the training area and the whole study area. It also depends on the study's aim and the study area's heterogeneity. The literature [***Triglia et al., 2013; Mohan et al., 2014***, Khanna, et al., 2021] finds that this percentage could range from 30 % to 100% approximately of the area. The most common is that the training area is about 70 % of the whole area. For instance, *Mondini et al., 2011* use a validation area of 30 % and a training area of 70%. In our work, having to test a new methodology, calibration and validation have been carried out on the entire 2016 area to provide a rigorous and precautionary approach. Undoubtedly, the manual delineation of landslides for calibrating an automatic procedure, such as the one proposed, is time-consuming. Still, it ensures a robust validation of the proposed methodology. In order to better clarify these aspects, we modify the sections "3.1.4 Parameters calibration based on ML inventory comparison" and "3.2.2. High-resolution images for ML inventory".**

**In the 2019 area, we applied the proposed methodology as in an ordinary scenario; therefore, we performed, over 10% of the area training and validation, and then, with the previously defined parameters, we applied the method to 90% of the area, producing the inventory.**

**To improve this Section, we added some literature dealing with training/validation and study area proportion throughout the manuscript (see section 3.1.4)**

- Also, if a training area that is just a part of the entire domain is selected, what characteristics should it have in terms of morphology, land-use and other properties?

It is also true that land-use and geological/geomorphological settings variability is to take into account also (e.g., Reichenbach et al., 2014). **Random training areas representative of the whole study area heterogeneity would be more robust than a single training one. In our cases, both training/validation areas cover 100% of the analysis area, and all variations are included. The area of training/validation in 2019 is also representative of the larger application area.**

**Reichenbach, P., Busca, C., Mondini, A.C. and Rossi, M., 2014. The influence of land use change on landslide susceptibility zonation: the Briga catchment test site (Messina, Italy). Environmental management, 54, pp.1372-1384.**

In line 251-255 authors explain that for a study area small training areas are selected while for the other case study the whole domain is used. If for the same study area, small training areas and then the whole domain are used, does the performance of the procedure change? By a sort of iteration, is it possible to define an ideal dimension of the training area? Is it possible to confirm this ideal dimension for the second area or it changes due to the different morphological and land-use conditions?

**R: Surely, the training area's size influences the performance. As reported above, the training area is usually about 70%, and the validation is 30%. Despite this, the performance depends on the heterogeneity of the training area where parameters are calibrated. However, as clarified above, we did not use a classical validation/training approach but a more precautionary one because we needed to test a new methodology.**

- In the study area Section, I suggest to add a climatological setting description particularly (but not only) focused on extremes frequency and its possible variation in recent years.

  **R: We improved Section 2 "Study area" (please see lines 105-110), adding a general climatological setting description associated with some references to the climatological study. In general, the literature and the data are not consolidated to understand the presence of solid trends in extreme rainfall events, especially for the scale of our study areas. In addition, several citations to meteorological events and their relation with climate are already reported in the description of single events (2.1 and 2.2) that we moved to the general introduction section.**

- Results should be discussed against previous literature e with major detail.

  **R: We improved the results sections and added references to compare our most relevant results with those from other studies.**

  **Specifically, we added the following paragraph in section 4.3.1:**

  **"The performance metrics Precision (P) P= TP/(TP+FP), Recall R= TP/(TP+FN) and F1-Score F1= 2TP / (2TP + FP + FN) without considering the intersection cases are reported in Table 4. The overall performance of our methodology is comparable with other studies, especially those using middle-resolution images (Ghorbanzadeh et al., 2021; Mondini et al., 2011; Handwerger et al., 2022). A recent study (Ganerød et al., 2023) using Sentinel-2 shows different performances depending on the study area in Norway; the best is from the deep-learning approach with a U-net architecture. Using high-resolution Planet images (Bhuyan et al., 2023) achieves high performance over several study areas. However, we assume that the different study/training/validation settings, the image used, and the event type made this comparison relative."**

**Table 4. performance metrics of this methodology.**

| AOI | P | R | F1 |
|---|---|---|---|
| Arroscia / Tanarello 2016 | 45 % | 64 % | 0.529 |
| Gavi 2019 | 69 % | 57 % | 0.623 |

- I suggest a general revision of English and a thorough formal proofreading as well.

  **R: We provided to improve English by several crosse re-reading by authors and the use of professional languages editor.**

**Detailed comments**

1. L12: it is the first step towards […] --> unclear **R1: We rewrite the sentence to clarify the meaning**
2. L17: hatted --> hit? **R2: Thank you, we corrected it.**
3. L19: well match --> match well. **R3: Thank you, we corrected it.**
4. L21: Keywords --> I suggest to use keywords that are not included in the title. **R4: Thank you for your observation; we change some keywords.**
5. L25: during flash floods […] shallow landslides --> what is the typical depth range of what you define shallow landslide? Also, flash floods and shallow landslides are two different phenomena both related to extreme rainfall events, they can happen simultaneously but I would not connect them directly. **R5: You are right. We incorrectly used the term "flash flood" instead of "extreme rainfall/events"; we corrected it in the manuscript. Compared to shallow landslides, this type of landslide in literature is defined as a small volume of earth featuring reduced thickness, commonly less than 2 m, triggered either by high-intensity rainfall or prolonged low-intensity one (c Guzzetti et al. 2004).**
6. L41: satellite images resolution nowadays is not so different […] --> which is the typical resolution? **R6: We better specify that most of the commercial satellites reach a sub-metric resolution that is comparable with aerial photo**
7. L43: what do you mean for dedicated acquisition planning? Aren't satellite orbits sort of fixed and so the revisiting time defined? **R7: You are right. The orbits are fixed, such as the revisit time. However, commercial HR satellites (e.g. the data used by Google Earth) usually frequently acquire images only over planned areas (e.g. reserved by their costumer) or in case of emergency planning programs (e.g., Copernicus EMS). It is also true that some new constellations of satellites (e.g., Planet) have a worldwide and constant revisit time.**
8. L54-61: I suggest to rewrite this paragraph making explicit the general and the specific objectives. In addition, emphasize the novelty in comparison to previous literature **R8: Thank you for your observation. The previous literature is generally resumed in lines 38-57, particularly in lines 50 – 55, where references to recent work that used GEE were added. At the same time, we rewrote the paragraph better to prove our approach's novelty and differences of our approach.**
9. L94: more steep slope in Serravalle Formation --> more steep slope than what**? R9: We clarified this sentence by modifying the manuscript. We mean that in the area where Serravalle Formation outcrops, the slopes are steeper than in the rest of the training area.**
10. L102: NW Alps have been affected --> always? Recently? Has the frequency changed with time? Some of this should be included in the climatological setting to be added in the study area section. **R10: In agreement with the previous comment, we add a**

**climatological setting description of the area of interest associated with some references to the climatological study. In general, this territory is usually affected by extreme rainfall events. There is evidence of a probable increase in present and future times; however, the literature and the data are not consolidated to understand the presence of solid trends in extreme rainfall events, especially for the scale of our study areas. In addition, several citations to meteorological events and their relation with climate are already reported in the description of single events (2.1 and 2.2) that we moved to the general introduction section.**

11. L105: Especially on a short time interval --> unclear, please quantify the short time interval. **R11: 24 hours**

12. L109: 650-700 mm --> in what time? Five days? **R12: Yes, in five days, we added it into the text**

13. L112: almost accurate --> what is it meant for almost? Which method was used? **R13: We change the sentence we mean a 1 km spatial resolution of rainfall map made with inverse distance weighted interpolation**

14. L129: intensity --> hourly intensity or instantaneous intensity? **R14: It is the hourly intensity, we corrected the manuscript**

15. L149-150: the difference between a map of areas most affected by landslides and a map of landslides is the difference between generally unstable areas and single landslides? **R15: thank you for your question that points out that we were not adequately exhaustive in the paragraph. We mean that the proposed methodology, the semi-automatic mapping (PL inventory), made with Sentinel-2 data, allows for identifying the area with likely high landslide density. Then, this area is where to focus the mapping using high-resolution data. We change the sentence in the manuscript.**

16. L154-157: please specify the resolution difference between satellite images and high-resolution images. Also, which is the source of high-res images? **R16: We used moderate-resolution satellite images (10 m Sentinel-2) and high-resolution (< 1m both from satellite Google Earth / Maxar and aerial type). See also Table 1**

17. L161: using slope and other geomorphological parameters --> In this phase I would say terrain and geomorphological properties. **R17: Thank you, corrected**

18. L175: is the same period of the year of this point ii more restrictive than the period June-September? **R18: Thank you for the comment that helped us to improve this point. These are two different restrictions, the first is needed to minimize the difference in the Sun angle between the two images, and it is a global rule. The second one help to define the period in which the NDVI contrast is enhanced and snow cover is limited; this is a local variable (depending on the climate zone considered). We change the sentences to make this constraint not limited to a specific climate zone.**

19. L181: averaged NDVI --> spatially averaged**? R19: No temporally averaged, we add it in the text**

20. L182: filtered by cloud cover --> 5%? **R20: Yes, we add the value**

21. L184: these constraints --> it is not very clear how the Novak algorithm takes into consideration the constraints listed in the bullet list I, ii, iii. **R21: We started from the Novak algorithm to learn how to calculate the NDVI on GEE. Then, we specifically added the constraints with the help of several tutorials available in the GEE user's forum. Here is an example of a GEE code to select summer images of 2020 < 3 %**

    a. // Create image collection of S-2 imagery for the period (T1-TX)
    b. var S2-post = ee.ImageCollection("COPERNICUS/S2")
    c. .filterDate("2020-06-01" , "2020-08-31")
    d. //and filter by cloud percentage
    e. .filter(ee.Filter.lt('CLOUDY_PIXEL_PERCENTAGE', 3));

22. L194: manually select NDVIvar threshold --> in a single image (raster) is the threshold the same or it can vary from area to area? **R22: In a single raster of NDVIᵥₐᵣ, the threshold is the same for all areas.**

23. L195-196: the whole sentence --> It is unclear, can you please provide an example (or a couple)? **R23: Thank you for the comment; we rewrite the sentence better. We also add an example from our case study. The figure below (not added in the manuscript) shows an example of NDVI time series made with GEE in which areas affected or not by landslides are compared. Based on a visual pattern of NDVIᵥₐᵣ, a threshold of -0.15 was chosen as optimal for the 2016 case study. Most shallow landslides show a decrease of 0.15 of NDVI (considering the summer season of 2018) from pre-event conditions. By contrast, the area not affected by landslides had almost the same value of NDVI during the period. The same was made for the 2019 case (Figure below )**

[Figure]

See on GEE: **https://code.earthengine.google.com/eabf52f1c93b65cd6f6f11365d968530**

[Figure]

See on GEE: https://code.earthengine.google.com/881a91cae51cc3a34bb89f811402f07d

24. L202: the value is empirically based[…] --> in this case the threshold is unique, right? **R24: Yes, the threshold is unique. We did not explicitly use a numerical value in the methodology because there is a range of variability as they are based on empirical observation.**

25. L203-204: additional filters maybe introduced --> such as? In addition, maybe based on what? It's a bit obscure. **R25: We add some examples to clarify this point; one is the removal of the shadow area (e.g., typically located at north of steep and high cliffs)**

**or in the areas that overlap with the main riverbed (to reduce the false positive related to river erosion processes)**

26. L209-210: The parameters used to […] --> was a formula developed? **R26: We mean the parameters (ray of interaction, cell size) to create the Kernel density map of PL. We did not develop a formula; we used a standard GIS tool (specifically in QGIS). A reference to the general Kernel density principle was added, too.**

27. L222: The iteration step aims […] --> when is the iteration stopped? Which is the threshold used to exit it? **R27: We did not correctly use the word iteration here. We replace it with "PARAMETERS CALIBRATION" because this is not an automatic process that runs in some software but a manual calibration.**

28. L239: The Boolean raster […] --> how is this obtained? With a single 10 m x 10 m cell there are four 5 m x 5 m cells. Did you use an average value? Majority? Else? **R28: In this case, the raster with the coarse resolution has the prevalence, and QGIS use the averaged values of the 4 cells of 5 m x 5 m, e.g., to match with 10 m x 10 m cell. We also try to use the 5 m x 5 m as resolution, but the results are almost the same**

29. L247: Table 1 --> I believe it could help the reader to have the vent date in the Table. **R29: Thank you, added.**

30. L271-272: algorithm based on Novak et al. (2021) --> is it exactly the same algorithm described in Section 3.1 or there are some differences? **R30: We renamed "algorithm" to code because it is the term more suitable. As described in the previous reply, we use this code as a "template" but we adapt this code to our aims with the help of several tutorials available on the GEE user forum**

31. L273: TS --> please define the acronym. **R31: Done. It is "time series."**

32. L274: estimate the recovery of vegetation --> if this is a specific objective of your study, please clarify it in the introduction. **R32: It is not a specific objective, it is a secondary aim which is helpful to understand the maximum time in which a post-event image is helpful for NDVIvar calculation (e.g. we note that the recovery for the 2019 case was faster than in 2016). That means the capacity to detect shallow landslide decays with vegetation recovery.**

33. L288: The characteristics of FP --> what characteristics? **R33: Mainly the distribution of NDVIvar and slope distributions, not only for FP but also for TP. Based on this, it is possible to calibrate thresholds: for instance, if 95 % of TP have slope > 17° we can set this value as a new threshold.**

34. L294: landslide dimension or land use --> why not using terrain properties such as aspect, slope, flow accumulation? **R34: We used terrain properties for the single intersection properties, such as TP and FP, as reported in the results section (e.g. 4.2.2 paragraph). Landslide dimension and land used are the main factors driving DR because they relate to satellite spatial resolution and the NDVI-based methodology.**

35. L296: Table 3 --> can you please clarify the difference between PP and PD? How did you distinguish between the two? Does it mean that in PP, PI is larger than ML while for PD is the opposite? **R35: We used the categories PP and PD related to the problem of partial intersection related to the pixel size of Sentinel-2 and the HR manual mapping. In other words, PP is a portion of PL not included in the intersection with ML, and vice versa, PD is a portion of ML not included in the intersection with PL. About the second question, no, it is just a matter of overlapping. We also added the figure below for a better understanding.**

[Figure]

a.

b. **A simple schema that shows the intersection case**

36. L305-308: The comparison […] characteristics --> methods not results. **R36: You are right. We removed this sentence**

37. L330: not filtered because the hydrographic network has no precise geocoding --> unclear. **R37: We mean that, due to the low spatial resolution of the available hydrographic network and its derived buffer, not all PLs are intersected despite on ground truth they are close to the riverbed**

38. L346-347: The intersection […] parameters --> methods not results. **R38. Thank you. We changed the sentence, and it was adapted to the results.**

39. L355: the methodology detects a landslide in 60% of the cases --> These TP-FP analysis results mean that the methods returns a over-representation of landslides. What about FN? Please discuss Fig 6C and 6D for this aspect as well. **R39: We specify in the text that "the methodology correctly detects a landslide in 60% of the cases" that means that 60% of PL are effectively landslides while the other 30% false positives. We add a sentence in which we remand to section 4.3 the statistic of FN/PD intersection cases. The TP/FP analysis reveals the limits of PL methods related to the parameters. In contrast, the TP/FN analysis shows the underestimation related to landslide size and the spatial resolution of the satellite. This is why we split the analysis into two separate sections and figures. To avoid confusion, we moved figures 6 and 10 before and split the paragraph into** "PL and ML intersection results" AND "PL validation statistics".

40. L377: Fig. 7 --> what about false negative? **R40: False negatives are not considered in these statistics because they correspond to manual landslides not detected by automatic mapping, and their parameters (slope or NDVI) are not helpful for calibration. Most of the false negatives are related to the small dimension of landslides compared to Sentinel-2 spatial resolution. This aspect is shown and discussed in Figure 12**

41. L386-387: we obtained the NDVIvar […] --> which was? **R41: Thank you, corrected.**

42. L387: Fig. 4D --> 8D? **R42: Thank you, corrected**

43. L400-401: It is possible […] manual mapping --> unclear. **R43: The sentence was removed because it is unnecessary.**

44. L404-405: the trigger points […] not detected --> This would not be a problem for an inventory intended for landslide susceptibility analysis, it maybe a problem for risk assessments. I suggest a brief discussion. **R44: This point is crucial because it explains one of the causes of the mismatch between manual and automatic mapping (especially in the shape of a landslide). At the same time, the fact that automatic mapping detects most of the landslides, even partially in shape, is positive. We also remember that the inventory is not intended only for landslide susceptibility or risk assessment. The main aim of the work is to verify the efficacy of automatic mapping to detect shallow landslides.**

45. L419: Fig 6A --> 10A? **R45: Thank you, corrected**

46. L427-428: The better performances […] 2019 event --> Are training areas absolutely comparable? **R46: They are almost comparable in terms of land use. The main differences are related to the events' intensity and the availability of pre- and post-event images. This is why performances are different.**
47. L454: it is possible to note […] small landslides --> unclear. **R47: Thank you, we better rewrite this sentence. We mean that the charts show that only a tiny fraction of small MLs are detected by PL (i.e., the red part of the bars).**
48. L476-477: The ML and PL […] case studies --> unclear, agree with each other? **R48: We rewrite this paragraph that is related to the further question (49)**
49. L479-489: whole paragraph --> These results are sort of calibration results (training and application areas are the same), correct? What if you apply the method outside the training area? This relates to my first general comment. **R49: This is more a sort of "validation and calibration" than a "calibration": the distribution of PL well matches the ML, which means the automatic methodology efficiently detects the most affected areas. To reply to your second question, we should have mapped manual landslides outside the training area. As reported in the methodology and in reply to your first comment, the large 2019 area is not validated: we assumed that is the first practical application of our PL methodology. We did not make such a time-consuming mapping because we already made a 1:1 training and validation in the small 2019 area; thus, the parameters used for the large 2019 area are already calibrated. However, manually and randomly checking on Google Earth (e.g., see figures below), we noticed the same performance outside the training area.**

[Figure]

[Figure]

**PL over 2019 large study area overlapped to Pre- and post- event images on Google Earth (N 44.6539, E 8.4715 ),**

50. L578: their density matches […] rainfall events --> unclear.: **R50: Thank you, we better rewrite this sentence.**
51. L582: landslides manually --> landslides manually detected? **R51: Thank you, corrected.**
52. L583: good agreement --> I believe it is worth mentioning that some parameters of the automatic recognition needed calibration. **R52: Thank you for the suggestions. We add the calibration of parameters as the fourth point of the methodology. It could be a general assumption (see also the flow chart of fig 3) not related only to this case study. The "good agreement" is assessed with the already calibrated PL.**
53. L584-585: whole sentence --> unclear, >60% of landslides with areas larger than 67 m2 (2/3 of 100 m2) but <20% for landslides smaller than 100 m2? **R53: Thank you. We better rewrite the sentence. We mean, "According to the findings, the semi-automatic method is capable of detecting the majority (about 60%) of shallow landslides larger than two or three times the size of Sentinel-2 ground pixels (100 m$^2$). In contrast, the PL method can identify only 20% of small landslides ( less than 100 m2).**
54. L595: for middle latitudes, the best comparison is with summer images --> This was an assumption in selecting the images (only June-September), I suggest not to present it as a conclusion. **R54: this assumption is based on the observation of the NDVI time series; the seasonal cycle of NDVI shows that the peak of vegetation activity in this climate area is in early summer. Thus it is also the period of enhanced NDVI contrast.**

---

## Author Response (AR2)

I found the manuscript of Notti et al much improved and I have appreciated the effort the authors put in answering my comments and specific questions. Before publication, I have just few additional detailed and minor comments (listed below), which I would like the authors to address. Line numbers refer to the manuscript with track-changes.

**R: The authors would thank reviewer 2 for his/her positive, rapid and accurate revision and for appreciating our effort to improve the manuscript. In this second revision, we further improve the manuscript according to her/his minor suggestions and reply point-by-point here following. The comments are also reported in the track-changes version of the new manuscript.**

1. Title, free Sentinel 2 data/images? **R1: Thank you for the suggestion; we added "images" to the title**

2. L11 and land management **R2: Thank you, corrected**

3. L12, I would clearly state :that: it is a two-step procedure, where the first step is "automatic" and based on the analysis of NDVI images to recognize potential landslide area, while the second step is manual and consists in mapping, within the potential areas, the actual landslides. **R3: Thank you for your suggestion, we added the statement, but with some changes: as in the flow chart, we propose the methodology made of several steps (3). As stated in the abstract and introduction, our purpose is made of 2 steps, which may generate confusion. So we used "phases" instead of "steps". Another point is that we used "semi-automatic" instead of "automatic" because the parameters used to create PL are empirically based, and manually set is not a fully automatic procedure like in machine learning work.**

   **It is also important to say that our primary focus is on the semi-automatic and fast detection of PL, while the second phase could be as a "comparison/validation" as made in this work, or the high-resolution mapping focused only where the PL map shows high density.**

4. L18 We apply this procedure to. **R4: thank you, corrected.**

5. L29, is the activation of shallow landslides (Gariano e Guzzetti, 2016) as defined in Guzzetti et al (2006) and Caine (1980). Remove unfortunately at the beginning of the next sentence. **R5: We corrected the references and removed "unfortunately."**

6. L170, same as for L12 **R6: We modify the text according to the reply R3**

7. L184, implementation of a filter using terrain and other geomorphological properties to obtain… **R7: Thank you, we corrected the mistakes in the sentence.**

8. L202, to obtain a strong contrast: **R8: Thak you, we corrected the typo**

9. L233, on G? > **R9: It is GEE (Google Earth Engine), we corrected it.**

10. L264, this might be a lexical issue but according to me validation refers to testing a model (an algorithm) on an unknown dataset. Calibration and validation cannot have the same (100%) target. **R10: Thank you for your suggestion. We agree that our procedure to check the data cannot be technically defined as a"validation" so we used only used calibration and a more generic "test".**

*11.* L290, in the answers to my comments you mentioned that slope was averaged for the 10 m cell, I would add it in the text. **R11: We added the following sentence to better explain also in the manuscript text:** "*..and adopted to the spatial resolution of Sentinel-2 in the raster calculator of QGIS (i.e. it averages the values of 4 cells of 5x5 m into one cell 10x10 m)."*

12. L370, the Figure you added in response to my comment n. 35 is incredibly clear to explain PP, PD etc. I would add it in the manuscript and Table 3 could be incorporated in the Figure .**R12: Thank you for your suggestion and for appreciating the graphical schema we provided to add a better figure that includes an improved description of intersection TABLE 3, as a legend of the figure**.

[Figure]

| Intersection type | Denomination | legend |
|---|---|---|
| Area of the full intersection of PL and ML | True Positive (TP) | |
| PL intersected by ML but not overlapping, i.e. the part of PL polygon not overlapping by ML is classified as PP while the overlapping part is a TP | Partial Positive (PP) | |
| ML intersected by PL but not overlapping, i.e. the part of ML polygon not overlapping by ML is classified as PD while the overlapping part is a TP | Partial Detection (PD) | |
| Detected by PL not Mapped in ML | False Postive (FP) | |
| Not Detected by PL and Mapped in ML | False Negative (FN) | |

**Figure 1: Graphical examples and the classification of the possible intersection cases of ML and PL inventory.**

13. L505, I recall my comment n. 44 and your answer. I agree that the main aim is to verify the efficacy of automatic mapping to detect shallow landslides, but as you stated in the introduction good inventories obtained quickly can improve damage estimations (i.e., risk assessment), susceptibility models and land management. Commenting on strong points and limits of the inventory you derive with your methodology for different applications (specifically two that are mentioned in the abstract and in the introduction) is interesting! It might be added in the conclusions. **R13: Thank you for the comment. We added some considerations in the conclusions, as you suggested: "*In summary, the PL inventory has strengths, such as its ability to: quickly map large areas, cost-effectiveness, and user-friendly processing. However, it also has limitations, including low resolution and a higher likelihood of false positives. Therefore, the primary purpose of the PL inventory is*

*not to generate precise susceptibility or damage assessment maps. Instead, it serves as an initial and swift assessment map, allowing users to identify the most heavily impacted areas. These areas can be targeted for further detailed mapping or on-site field surveys"*. **About the application of this methodology, we recently made a rapid mapping of the effect of extreme rainfall events that hit the Emilia-Romagna region last May (here is the preliminary map [https://zenodo.org/record/7995624](https://zenodo.org/record/7995624)) that allowed us to find the area most affected.**

14. L606, effectiveness instead of efficiency? **R14: You are right; "effectiveness" better matches the context, changed.**